# Words & Weights: Streamlining Multi-Turn Interactions via Co-Adaptation

**Chenxing Wei** [1 2 3]  **Hong Wang** [4]  **Ying He** [1]  **Zhongxiang Dai** [5]  **Bo Jiang** [3]  **F. Richard Yu** [6]  **Yao Shu** [7]

## Abstract

Test-time policy adaptation for multi-turn interactions (T²PAM) is essential for aligning Large Language Models (LLMs) with dynamic user needs. However, existing paradigms typically treat adaptation as a single-axis problem by either purely refining instructions or solely updating weights. This bifurcated approach overlooks the fact that interaction failures arise from a coupled mixture of context ambiguity and model incapacity. To address this, we propose ROSA2, a framework that reformulates T²PAM as a joint optimization problem over the heterogeneous space of Words and Weights. Within this framework, the semantic stream acts as a feedback normalizer that transforms noisy or ambiguous user feedback into actionable instructions, ensuring that parametric adaptation is performed on semantically clarified trajectories. Theoretically, we prove that this semantic pre-conditioning strictly reduces the required parameter shift for convergence. Empirically, ROSA2 consistently outperforms state-of-the-art baselines across diverse mathematical, general reasoning, and coding benchmarks. It achieves up to a 37.8% accuracy improvement while reducing average interaction turns by 40%, demonstrating that ROSA2 unlocks the true potential of parameter updates.

## 1. Introduction

*Large Language Models* (LLMs) have demonstrated remarkable capabilities across a wide array of general-purpose tasks (Yang et al., 2025; OpenAI, 2025; Google, 2025), increasingly serving as collaborative agents that engage in complex, multi-turn dialogues to solve open-ended problems (Yi et al., 2025). However, a fundamental mismatch persists between static alignment paradigms (e.g., SFT (Ouyang et al., 2022; Wei et al., 2025a), RLHF (Shao et al., 2024; Wei et al., 2025c)) and the dynamic nature of real-world deployments (Li et al., 2025b; Laban et al., 2025). Consequently, pre-trained models frequently struggle during extended interactions (Wang et al., 2024). They exhibit limited adaptability (Yi et al., 2025) and poor self-correction capabilities (Deshpande et al., 2025), which ultimately leads to the performance stagnation documented in recent literature (Wei et al., 2025b). To bridge this gap without the prohibitive costs of continuous retraining, *Test-Time Policy Adaptation for Multi-Turn Interactions* (T²PAM) (Wei et al., 2025b) has emerged as a critical paradigm. T²PAM dynamically optimizes the model policy during ongoing sessions to ensure continuous alignment with user preferences, thereby significantly enhancing response accuracy.

Despite the significant promise of T²PAM, existing paradigms predominantly treat test-time adaptation as a *single-axis problem*. They typically resort to either purely refining instructions (*Prompt Engineering*) (Yi et al., 2025) or exclusively adjusting model weights (*Test-Time Training*), as demonstrated by representative approaches like ROSA (Wei et al., 2025b) and TTRL (Zuo et al., 2025). We challenge this bifurcated perspective by explicitly modeling the effective policy of an LLM as a coupled function $\pi(x, \theta)$ that depends simultaneously on its internal parameters (Weights) and the external context (Words). Conditional optimization strategies, which update one variable while freezing the other, overlook a fundamental reality: interaction failures stem from a complex interplay of *context ambiguity* and *model incapacity* (Keluskar et al., 2024). Addressing these factors in isolation has proven insufficient. Parameter-centric methods risk overfitting to noisy interaction histories, whereas prompt-centric methods frequently encounter intrinsic capability ceilings. This methodological misalignment ultimately harms downstream performance, resulting in low accuracy and unnecessarily prolonged interaction (Tang et al., 2025a). A comprehensive discussion of related work is provided in Appendix A.

To overcome these limitations, effective adaptation requires resolving a fundamental error attribution dilemma:

[1] Shenzhen University [2] Guangdong Laboratory of Artificial Intelligence and Digital Economy (SZ) [3] Bytedance [4] University of Science and Technology of China [5] The Chinese University of Hong Kong (Shenzhen) [6] Carleton University [7] Hong Kong University of Science and Technology (Guangzhou). Correspondence to: Yao Shu <yaoshu@hkust-gz.edu.cn>.

*Proceedings of the 43ʳᵈ International Conference on Machine Learning*, Seoul, South Korea. PMLR 306, 2026.

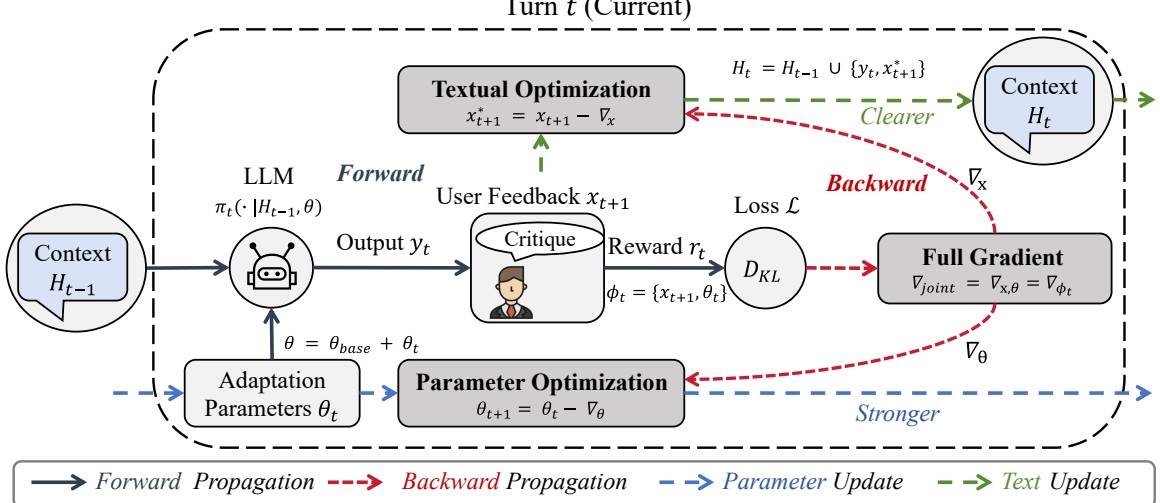

*Figure 1.* **Overview of the ROSA2 Framework.** We formulate T²PAM as a joint optimization over the coupled variables $\phi_t = \{x_{t+1}, \theta_t\}$. During the *Forward Phase* (solid lines), the model generates a response $y_t$ conditioned on history $H_{t-1}$. The *Backward Phase* (dashed lines) approximates the full gradient $\nabla_{\text{joint}}$ of the interaction loss $\mathcal{L}$ through two synergistic modules. **Textual Optimization** (top, green) uses textual gradients ($\nabla_x$) to refine user feedback into a clarified instruction ($x_{t+1} \rightarrow x_{t+1}^*$), resolving context ambiguity. Simultaneously, **Parameter Optimization** (bottom, blue) applies gradient updates ($\nabla_\theta$) to adjust adapter weights ($\theta_t \rightarrow \theta_{t+1}$), enhancing the intrinsic capability of the model. This co-adaptation ensures the system is both "Clearer" and "Stronger" for the next turn.

*When a model fails in a multi-turn context, is it due to a lack of intrinsic capability (parameter misalignment) or a misunderstanding of the task intent (context ambiguity)?*

Addressing these factors in isolation proves insufficient (Chen et al., 2025). Pure prompt engineering cannot remedy intrinsic capability deficits (Lee et al., 2025), whereas pure parameter adaptation is prone to learning spurious mappings from noisy inputs (Li et al., 2025a). As visualized in Figure 2(b), the optimization landscape of T²PAM is characterized by coupled semantic and parametric gaps. Approaching this coupled system via independent updates, which is analogous to following *partial derivatives*, leads to convergence at suboptimal local minima. Specifically, solely optimizing parameters gravitates towards an *Overfitting Trap*, while solely refining the context stalls in a *Deficit Trap*. Consequently, we posit that T²PAM must be reformulated as a joint optimization problem.

Crucially, we argue that these optimization paths are not merely additive but profoundly synergistic. We establish that **semantic clarity acts as a *pre-conditioner* for parametric alignment**. By prioritizing the elimination of semantic ambiguity, we effectively cleanse the learning signal. This ensures that the gradient descent for parameters is strictly oriented towards the true task intent rather than fitting accumulated noise. Furthermore, the Words stream functions as more than a conventional prompt optimizer for well-defined task instructions. In realistic multi-turn interactions, user feedback is frequently vague, noisy, incomplete, or even partially incorrect. ROSA2 treats such

feedback as raw observational evidence rather than a direct optimization target. The semantic stream distills the entire interaction history into a refined instruction alongside an internal adaptation signal. This normalization process ensures that subsequent parameter updates are computed on semantically clarified trajectories instead of raw, noisy prompts. This design enables ROSA2 to operate robustly in generalized open-ended dialogue settings where explicit ground-truth rewards are often unavailable. Qualitative examples demonstrating this feedback canonicalization process, along with open-ended dialogue evaluations, are provided in Appendix E. This perspective also aligns with recent research on model alignment (Liu et al., 2023; Bo et al., 2026).

Driven by this insight, we introduce ROSA2, a unified framework designed to approximate the full gradient of the interaction objective by co-adapting the semantic context and model parameters. Instead of treating error signals as a monolith, our approach effectively disentangles the optimization process. It employs textual gradients to sharpen the user intent (Words) and utilizes closed-form updates to enhance the intrinsic execution capabilities of the model (Weights). Theoretically, we demonstrate that this semantic pre-conditioning is rigorous, proving that it strictly bounds the magnitude of parameter shifts required to reach the optimal policy. This theoretical advantage translates directly into substantial empirical gains. ROSA2 establishes a new state-of-the-art across multiple benchmarks, achieving an accuracy improvement of up to 37.8% while simultaneously cutting interaction costs by reducing average turns by 40%. These results validate our core hypothesis: accurate context

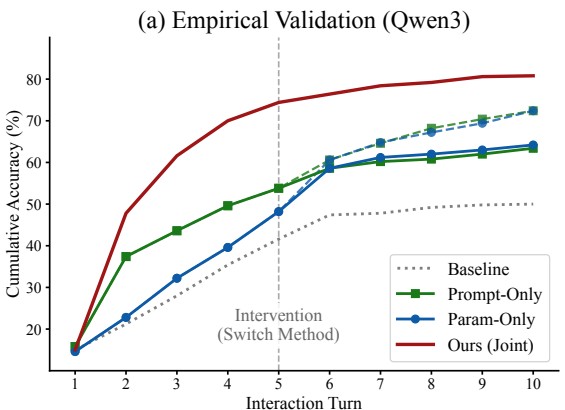

(a) Empirical Validation (Qwen3)

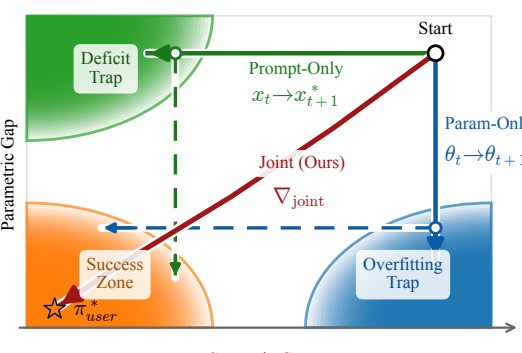

(b) Optimization Landscape

*Figure 2.* **Empirical Observations and Theoretical Landscape. Figure (a)** In the experimental results on MATH (Qwen3-8B) reveal that single-axis methods (**Green**/**Blue** solid lines) suffer from premature stagnation. However, the immediate recovery observed in the Switch experiments (**Green**/**Blue** dashed lines) suggests this bottleneck is structural. **Figure (b)** We map these dynamics to the optimization landscape using **consistent color and line styling**: the **Prompt-Only** path (**Green**) stalls in the *Deficit Trap* (Hitting capability ceilings), while the **Param-Only** path (**Blue**) gravitates towards the *Overfitting Trap* (Memorizing noise). The dashed arrows in Figure (b) visualize how the Switch Method escapes these local minima by activating the missing axis. Crucially, ROSA2 (**Red**) approximates the joint gradient $\nabla_{joint}$, forming an **Optimal Trajectory** that bypasses these traps and proceeds directly to the *Success Zone*, corresponding to the superior convergence shown in Figure (a).

is crucial for maximizing the effectiveness of parameter adaptation. Our contributions are summarized as follows:

- We propose ROSA2, a joint optimization framework over context and parameters that resolves the error attribution dilemma. Its Words stream normalizes noisy feedback into refined instructions for robust adaptation (Section 3).
- We prove that semantic pre-conditioning strictly reduces parameter shift (Theorem 4.1) and guarantees faster convergence (Theorem 4.2) (Section 4).
- Empirically, ROSA2 achieves up to a 37.8% accuracy gain and a 40% turn reduction with negligible overhead. Comprehensive ablations are in the Appendix (Section 5).

## 2. Motivation: The Traps of Conditional Optimization

T²PAM inherently requires joint optimization over the external context $x$ (Words) and model weights $\theta$ (Weights). We demonstrate that conditionally optimizing either dimension in isolation inevitably traps the model in suboptimal states, manifesting as either persistent reasoning deficits or prompt overfitting. The following subsections empirically and theoretically validate this conclusion to motivate our joint co-adaptation framework.

### 2.1. Experimental Setup

To empirically validate this hypothesis, we conducted a controlled study using the `Qwen3-8B` model on the `MATH` dataset (Hendrycks et al., 2021b) to simulate a challenging 10-turn interaction scenario. We compared four distinct optimization settings to isolate the effects of different variables. First, *Standard Inference* evaluates the model

performing multi-turn reasoning with both the prompt and model parameters frozen. Second, *Prompt Optimization* freezes the model parameters and exclusively updates the system prompt using TextGrad. Third, *Parameter Optimization* fixes the system prompt and exclusively updates the model parameters via ROSA. Fourth, the *Switch Method* tests the limitations of conditional optimization methods by implementing a crossover intervention at the observed stagnation point, specifically Turn 5. For the model initially optimizing prompts, we freeze the prompt and switch to updating parameters. Conversely, for the model initially optimizing parameters, we freeze the weights and switch to updating the prompt. Consequently, the Switch Method serves as a decoupled pipeline baseline. It tests whether a simple sequential combination of the two optimization axes can recover the behavior of continuous co-adaptation.

### 2.2. Observation: Stagnation and Recovery

The empirical results illustrated in Figure 2(a) demonstrate a notable trend. The Baseline curve exhibits limited self-correction capability and remains nearly flat. Furthermore, conditional optimization methods suffer from diminishing returns and eventual premature stagnation despite their initial gains. Specifically, the Prompt-Only method is constrained by *policy misalignment*, where semantic updates alone fail to bridge the intrinsic reasoning gap. Concurrently, the Param-Only method plateaus early due to *overfitting*. Crucially, a turning point occurs upon intervention. Implementing the *Switch Method* at Turn 5 triggers a distinct performance improvement. This rapid recovery indicates that the preceding stagnation was fundamentally driven by the structural limitations of conditional optimization.

## 2.3. Theoretical Landscape and Joint Optimization

We map these empirical results to the theoretical optimization landscape in Figure 2(b) to identify two distinct failure modes inherent to conditional updates. Specifically, the *Prompt-Only* method stalls in a *Deficit Trap* because purely semantic updates cannot rectify intrinsic reasoning deficits. Conversely, the *Param-Only* method falls into an *Overfitting Trap* because parameter updates easily overfit to ambiguous prompts without prior context refinement. The Switch experiments validate that escaping these suboptimal local minima strictly requires the presence of both semantic clarity and parametric capability. Building on this insight, we propose ROSA2 to implement a joint optimization strategy. Unlike the sequential Switch intervention, ROSA2 simultaneously updates both the semantic and parametric states after each failed interaction to prepare for the subsequent turn. By approximating the effect of the joint gradient of the interaction objective from the very first turn, ROSA2 leverages the complementary strengths of both dimensions to systematically bypass the aforementioned traps. As depicted in Figure 2(a), it follows an *Optimal Trajectory* (solid red line) that achieves significantly faster convergence and higher overall accuracy. Driven by this theoretical foundation, the subsequent section details the mathematical formulation and architecture of the ROSA2 framework.

## 3. Joint Optimization via Full-Gradient Approximation

Building upon the motivation established in Section 2, we propose ROSA2. This novel framework treats T$^2$PAM strictly as a joint optimization problem. By viewing the effective policy as a coupled function of *Words* (the external context) and *Weights* (the internal parameters), ROSA2 approximates the effect of the joint gradient of the interaction objective through trajectory-coupled updates. This mechanism seamlessly aligns the model policy with the latent optimal user preference.

### 3.1. Problem Formulation: Joint Optimization in the Current Turn

As illustrated in Figure 1, we consider the $t$-th turn of a multi-turn interaction session. Let $H_{t-1} = \{(x_1, y_1), \ldots, (x_{t-1}^*, y_{t-1}), x_t^*\}$ denote the immutable interaction history accumulated prior to generating the current response. This history contains the completed dialogue pairs from all previous turns alongside the refined query $x_t^*$ prepared for the current turn. At the current turn $t$, the model operates using the composed parameters $\theta = \theta_{\text{base}} + \theta_t$, where $\theta_t$ represents the current learnable adapter weights. The system generates the response $y_t$ according to the current policy $\pi_\theta$ conditioned on the accumulated history:

$$y_t \sim \pi_t(\cdot \mid H_{t-1}, \theta). \quad (1)$$

Subsequently, the user provides feedback denoted as $x_{t+1}$, which serves as the raw query for the subsequent turn. This feedback need not be a clean scalar reward. In open-ended interactions, it is frequently vague, noisy, incomplete, or even partially incorrect. Distinct from standard paradigms, we treat this feedback $x_{t+1}$ as an optimizable variable alongside the model parameters $\theta_t$. Specifically, the semantic stream canonicalizes the raw feedback into a refined instruction $x_{t+1}^*$ and an internal adaptation signal. This normalization allows the parametric stream to compute updates based on semantically clarified trajectories. Consequently, we define the set of joint optimization variables for the current step as $\phi_t = \{x_{t+1}, \theta_t\}$.

**The Joint Optimal Policy Construction.** We postulate the existence of a *Joint Optimal Policy $\pi^*$* that represents the ideal response distribution for the current interaction turn. Following the established principles of reward-weighted regression (Rafailov et al., 2023), we construct this target distribution by re-weighting the policy from the previous turn, denoted as $\pi_{t-1}$. In our co-adaptation setting, $\pi_{t-1}$ consistently serves as the reference policy for the current adaptation step (Wei et al., 2025b). Formally, we define this as:

$$\pi_t^*(y \mid H_{t-1}) \triangleq \frac{1}{Z_t} \pi_{t-1}(y \mid H_{t-1}) \exp\left(\frac{r(y)}{\beta}\right), \quad (2)$$

where $r(y)$ represents the internal adaptation signal derived from the user feedback and the task environment. Crucially, the partition function $Z_t$ depends exclusively on the previous policy $\pi_{t-1}$ and the fixed interaction history:

$$Z_t = \mathbb{E}_{y \sim \pi_{t-1}}\left[\exp\left(\frac{r(y)}{\beta}\right)\right]. \quad (3)$$

Therefore, $Z_t$ acts as a *constant scalar* with respect to the current joint optimization variables $\phi_t = \{x_{t+1}, \theta_t\}$.

**Optimization Objective.** Our primary objective is to update the current policy $\pi_t$, which is parameterized by both $x$ and $\theta$, to approximate this optimal target $\pi_t^*$. We mathematically formulate this goal as minimizing the *Forward KL Divergence*, denoted as the interaction loss function $\mathcal{L}$:

$$\mathcal{L}(\phi_t) = D_{KL}\left(\pi_t^*(\cdot \mid \phi_t) \,\big\|\, \pi_t(\cdot \mid \phi_t)\right). \quad (4)$$

Expanding the KL divergence yields:

$$\mathcal{L}(\phi_t) = \underbrace{\mathbb{E}_{y \sim \pi_t^*}[\log \pi_t^*(y)]}_{-E(\pi_t^*)} - \mathbb{E}_{y \sim \pi_t^*}[\log \pi_t(y \mid \phi_t)]. \quad (5)$$

The target policy $\pi_t^*$ is strictly fixed by the forward pass, as it is determined solely by $\pi_{t-1}$ and the signal $r$. Thus, its entropy term $E(\pi_t^*)$ remains entirely independent of the optimizable variables $\phi_t$. Consequently, minimizing this divergence is mathematically equivalent to minimizing the cross-entropy, which equates to maximizing the expected log-likelihood of the optimal policy:

$$\mathcal{L}(\phi_t) \cong -\mathbb{E}_{y \sim \pi^*}\left[\log \pi_t(y \mid \phi_t)\right]. \tag{6}$$

**Total Derivative and Co-Adaptation.** To execute this update, we analyze the *total differential* $d\mathcal{L}$ with respect to the joint variables $\phi_t$. We employ importance sampling to estimate the gradient expectation under the policy distribution $\pi_t$:

$$\nabla_{\phi_t}\mathcal{L} = -\mathbb{E}_{y \sim \pi_t}\left[\frac{\pi^*(y)}{\pi_t(y)}\nabla_{\phi_t}\log \pi_t(y \mid \phi_t)\right]$$
$$= -\mathbb{E}_{y \sim \pi_t}\left[\frac{1}{Z_t}\exp\left(\frac{r(y)}{\beta}\right)\nabla_{\phi_t}\log \pi_t(y \mid \phi_t)\right]. \tag{7}$$

Expanding the gradient operator $\nabla_{\phi_t}$ explicitly reveals the highly coupled nature of this optimization process. To strictly decrease the divergence, the total change in the loss function must follow the full gradient within the joint space:

$$d\mathcal{L} \propto$$
$$-\frac{1}{Z_t}\mathbb{E}_{y \sim \pi_t}\left[\underbrace{\exp\left(\frac{r(y)}{\beta}\right)}_{\text{Reward Weight}}\left(\underbrace{\nabla_x \log \pi_t \cdot dx}_{\text{Optimizing Prompt}} + \underbrace{\nabla_\theta \log \pi_t \cdot d\theta}_{\text{Optimizing Params}}\right)\right]. \tag{8}$$

Equation 8 theoretically mandates the necessity of joint co-adaptation for T$^2$PAM. Since $Z_t$ operates as a constant scaling factor derived from the prior turn, approximating the joint optimal policy strictly requires rectifying the query context $x$ and updating the internal parameters $\theta$ along the direction of the reward-weighted log-likelihood. Operationally, ROSA2 realizes this objective through cross-turn pre-conditioning. By enforcing this mechanism, interaction trajectories generated under refined Words provide significantly cleaner evidence for all subsequent Weight updates.

### 3.2. The ROSA2 Algorithm

Guided by the total differential derivation in Equation 8, we propose ROSA2, a co-adaptation framework designed to iteratively approximate the joint optimal policy throughout multi-turn interactions. The complete protocol is formalized in Algorithm 1. The process begins by initializing the turn counter $t = 1$, setting the learnable adapter parameters $\theta_1$ to zero, and establishing the current interaction history $H$ with the initial user query $x_1$ (lines 1-2 in Algorithm 1). At each turn $t$, the workflow executes two distinct phases:

**Phase 1: Generation and Evaluation.** To leverage the previously adapted knowledge, the system constructs the effective model parameters $\theta$ by adding the current adapter weights $\theta_t$ to the frozen base model parameters $\theta_{\text{base}}$ (line 5). A response $\hat{y}_t$ is subsequently generated using the current policy $\pi_\theta$ conditioned on the accumulated history $H$ (line 6). Following generation, the system receives an internal adaptation signal $r_t$ alongside user feedback intended for the next turn, denoted as $x_{t+1}$ (line 7). If the response is accepted ($r_t = +1$) or the maximum turn limit $T_{\max}$ is reached, the process terminates and returns the successful response $\hat{y}_t$ (lines 8-9).

**Phase 2: Joint Optimization.** If the response is rejected ($r_t = -1$) and the session continues, ROSA2 triggers the co-adaptation process to jointly optimize the system state for the next interaction. First, the *Semantic Stream* activates to resolve context ambiguity. It utilizes the specific deficiencies detected in the current response $\hat{y}_t$ to compute a textual gradient. This gradient is then applied to refine the raw incoming feedback $x_{t+1}$ into a precise and actionable query $x_{t+1}^*$ (lines 12-14). Uniquely, even if explicit user feedback is completely absent (i.e., $x_{t+1} = \emptyset$), this stream autonomously synthesizes a corrective instruction based on the error signal derived from the failure in $\hat{y}_t$. If the provided feedback is vague or noisy, the semantic stream processes it strictly as raw observational evidence. It canonicalizes this noise into a rigorously structured instruction for the subsequent turn. This mechanism guarantees that the model consistently receives a semantically optimized prompt regardless of the actual user guidance quality. By systematically generating this fine-grained and normalized feedback in every iteration, the system effectively bridges the semantic gap between the true intent of the user and the execution boundary of the model.

Simultaneously, the *Parametric Stream* utilizes the adaptation signal ($r_t$) and the current policy $\pi_\theta$ to estimate the latent target policy of the user, denoted as $\pi^*$. It subsequently computes a parameter update $\Delta\theta_t$ to align the model policy $\pi_t$ more closely with this estimated target $\pi^*$ (lines 15-17). This parameter update is computed directly from the current failure trajectory, which was generated under the refined Words established in previous turns. Consequently, Words effectively pre-condition Weights throughout the multi-turn trajectory. Meanwhile, the current textual and parametric updates can be executed entirely in parallel to prepare $x_{t+1}^*$ and $\theta_{t+1}$ for the subsequent response. The computational efficiency of this parallel, one-step update mechanism makes it highly suitable for real-time multi-turn interactions. It allows for rapid iterative updates that continuously align the model policy with the evolving preferences of the user.

Finally, the system prepares for the next iteration by updating the interaction history $H$. This update incorporates

---

**Algorithm 1** ROSA2 Co-Adaptation Protocol

---

1: **Input:** Initial Query $x_1$, Base Model Parameters $\theta_{\text{base}}$, Max Turns $T_{\max}$.
2: **Initialize:** Turn Counter $t \leftarrow 1$, Adaptation Parameters $\theta_1 \leftarrow \mathbf{0}$, Current History $H_0 \leftarrow \{x_1\}$.
3: **while** $t \leq T_{\max}$ **do**
4:     // **Phase 1: Generation and Evaluation**
5:     Compose parameters: $\theta \leftarrow \theta_{\text{base}} + \theta_t$.
6:     Generate response: $\hat{y}_t \sim \pi(\cdot \mid H_{t-1}, \theta)$.
7:     Receive reward $r_t$ and feedback $x_{t+1}$ (next turn query) from Environment/User.
8:     **if** $r_t = +1$ **or** $t = T_{\max}$ **then**
9:         **Return** $\hat{y}_t$ // Task completed or limit reached
10:    **end if**
11:    // **Phase 2: Joint Optimization**
12:    // *Step A: Semantic Update (TextGrad)*
13:    Compute semantic gradient and refine query:
14:     $x_{t+1}^* \leftarrow x_{t+1} - \nabla_{\text{text}}\mathcal{L}(\hat{y}_t)$
15:    // *Step B: Parametric Update (ROSA)*
16:    Construct target distribution $\pi^*$ using $\pi_\theta$ and $r_t$.
17:     $\theta_{t+1} \leftarrow \theta_t - \nabla_\theta\mathcal{L}(\theta \mid r_t, \pi^*, \pi_\theta)$
18:    Update History: $H_t \leftarrow H_{t-1} \cup \{\hat{y}_t, x_{t+1}^*\}$
19:    $t \leftarrow t + 1$
20: **end while**

---

both the current response $\hat{y}_t$ and the *refined* query $x_{t+1}^*$, thereby ensuring that all subsequent generations are strictly conditioned on the optimized context (lines 19-20).

**Advantages.** The ROSA2 framework provides a robust solution to T²PAM that is explicitly derived from the joint-gradient approximation. By co-adapting both the semantic context and the model parameters, it systematically overcomes the intrinsic limitations of conditional optimization baselines. Specifically, the *Semantic Stream* normalizes raw feedback into clear, actionable instructions. This mechanism effectively addresses scenarios where explicit feedback is absent, ambiguous, or highly noisy. Complementarily, the *Parametric Stream* ensures that the model possesses the intrinsic capability required to execute these refined instructions. This synergistic loop enables ROSA2 to robustly handle ambiguous inputs and autonomously recover from errors. As a result, it significantly improves the overall success rate in complex, multi-turn tasks. Additional open-ended evaluations and qualitative examples detailing this feedback normalization process are provided in Appendix E.

## 4. Theoretical Results

Building upon the joint optimization formulation established in Section 3.1, we now establish the convergence properties of the ROSA2 framework. Specifically, we analyze how the joint updates of the semantic query $x$ and the model pa-

rameters $\theta$ (Equation 8) theoretically drive the model policy toward the latent optimal user policy $\pi_{\text{user}}^*$. This theoretical analysis proceeds in two distinct stages. First, we examine the *mechanistic synergy* in Section 4.1. We prove that semantic refinement strictly reduces the norm of the required parameter shift (Theorem 4.1). Subsequently, we extend this local property to a global perspective in Section 4.2. We derive a unified convergence bound (Theorem 4.2) that explicitly quantifies the reduction in divergence from the optimal user policy while rigorously accounting for approximation errors.

### 4.1. Mechanism: Parametric Error Reduction

We first analyze the impact of optimizing the context $\mathbf{x}$ on the parametric optimization of $\theta$. A fundamental insight is that refining the context $\mathbf{x}$ significantly reduces the magnitude of the parameter shifts required to achieve alignment. We formalize this phenomenon in the following theorem.

**Theorem 4.1** (Reduction of Parameter Shift). *Let $\Delta\theta_t(\mathbf{x})$ denote the solution to the linearized parameter update defined in Equation (6) of ROSA (Wei et al., 2025b) given a query $\mathbf{x}$. If we successfully update the query from $\mathbf{x}_t$ to $\mathbf{x}_t^*$ such that the semantic gap to the user intent is reduced (i.e., $D_{KL}(\pi_{\text{user}}^*\|\pi(\cdot|\mathbf{x}_t^*)) < D_{KL}(\pi_{\text{user}}^*\|\pi(\cdot|\mathbf{x}_t))$), then the norm of the required parameter update satisfies:*

$$\|\Delta\theta_t(\mathbf{x}_t^*)\|_2 < \|\Delta\theta_t(\mathbf{x}_t)\|_2 \tag{9}$$

**Remark.** The detailed mathematical proof is provided in Section B.1. Theorem 4.1 underscores the synergistic necessity of co-adapting both $\mathbf{x}$ and $\theta$. By aligning the input context with the existing knowledge boundary of the model first, we minimize the residual error that the parameters must ultimately correct.

**Empirical Evidence.** Experimental observations illustrated in Figure 3 strongly corroborate this mechanism. The parametric error of ROSA2 (blue line, $\|\Delta\theta\|^2$) is significantly reduced compared to the ROSA baseline (gray line). This structural reduction confirms that semantic refinement strictly decreases the optimization difficulty for the parametric stream.

### 4.2. Unified Convergence Bound

Building upon Theorem 4.1, we derive a unified bound that quantifies the overall performance of the co-adaptation framework. This theorem extends Theorem 4 from (Wei et al., 2025b) by explicitly accounting for the approximation errors in both the semantic and parametric spaces.

**Theorem 4.2** (Unified Convergence Bound). *Assume that the local second-order expansion of the KL divergence is smooth over the Cartesian product of the token-embedding space and the adapter-parameter space. Let the block Hes-*

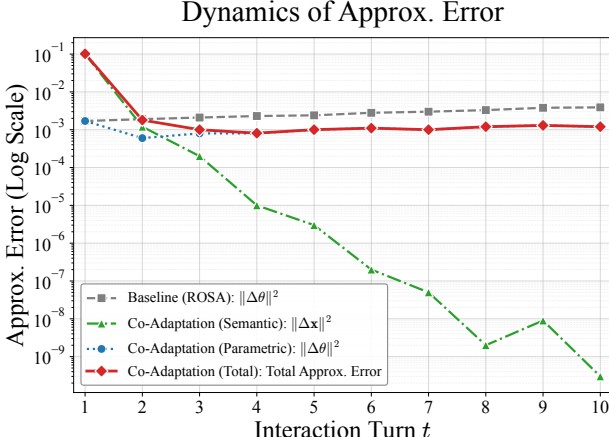

*Figure 3.* Dynamics of approximation error terms on MATH using Qwen3-8B. The semantic update magnitude $\|\Delta \mathbf{x}_t\|_2^2$ is computed as an $L_2$ distance in the continuous token-embedding space. The parametric curve of ROSA2 and the ROSA baseline start from the same adapter initialization at Turn 1. The total error includes both semantic and parametric components, so it can start higher due to the non-zero initial semantic discrepancy.

*sian spectral norms be bounded by $L_x$, $L_\theta$, and $L_{x\theta}$, and define $\bar{L} = \max(L_x + L_{x\theta}, L_\theta + L_{x\theta})$. After $T$ turns of co-adaptation, the divergence between the final policy $\pi_{\phi_T}$ and the optimal user policy $\pi^*_{\text{user}}$ is bounded by:*

$$
D_{\text{KL}}(\pi^*_{\text{user}}\|\pi_{\phi_T}) \leq \underbrace{D_{\text{KL}}(\pi^*_{\text{user}}\|\pi_{\phi_0})}_{\text{Initial Error}}
$$

$$
\underbrace{- C \sum_{t=1}^{T} \pi_{\theta_{t-1}}(\mathbf{y}_t|\mathbf{x}_t^*)}_{\text{Improvement}} + \underbrace{\frac{\bar{L}}{2} \sum_{t=1}^{T} \left( \|\Delta \mathbf{x}_t\|_2^2 + \|\Delta \theta_t\|_2^2 \right)}_{\text{Approx. Error}} .
$$

$$(10)$$

*where the scalar $C > 0$ absorbs the reward-weighting constant. Furthermore, the magnitude $\|\Delta \mathbf{x}_t\|_2^2$ is computed strictly within the continuous token-embedding space, and $\|\Delta \theta_t\|_2^2$ is evaluated within the adapter-parameter space.*

**Remark.** The detailed mathematical proof is provided in Section B.2. Theorem 4.2 formally decomposes the convergence dynamics into three interconnected components. First, the *Initial Error* serves as the constant baseline divergence at the beginning of the interaction. Second, the *Improvement* term quantifies the cumulative error reduction driven by user feedback. Crucially, the joint adaptation process amplifies this term by refining the query into an optimal form $\mathbf{x}_t^*$. This refinement ensures that the model generates responses $\mathbf{y}_t$ with significantly higher probability mass under the latent optimal user policy $\pi^*_{\text{user}}$. Finally, the *Approx. Error* reflects the penalty incurred from inexact iterative updates. Although ROSA2 introduces an additional semantic cost $\|\Delta \mathbf{x}_t\|_2^2$, it systematically mitigates the total

error through the mechanism established in Theorem 4.1. This mitigation is visually corroborated by the red line in Figure 3.

**Empirical Evidence.** As illustrated in Figure 3, the query context $\mathbf{x}_t$ progressively approaches the optimal form $\mathbf{x}^*$. Correspondingly, the squared norm of the semantic update $\|\Delta \mathbf{x}_t\|_2^2$ exhibits an exponential decay, which is depicted by the green line. Consequently, the total approximation error of ROSA2 is initially high due to the large initial semantic discrepancy. However, this error rapidly drops and remains significantly lower than that of the isolated baseline method, as shown by the red and gray lines respectively. This rapid stabilization empirically validates that ROSA2 achieves a strictly lower overall approximation error throughout the interaction trajectory.

## 5. Empirical Results

Following the protocol established in Section 2.1, we employ an automated evaluation pipeline across verifiable benchmarks, where correctness is rigorously validated via ground-truth matching for reasoning tasks or execution-based unit tests for coding and agentic tasks. This controlled setting provides highly reproducible feedback signals. Interactions persist until the model succeeds or reaches a predefined turn limit, allowing for the simultaneous measurement of efficacy (success rate) and efficiency (turn count). Comprehensive details regarding the experimental setup and hyperparameter configurations are deferred to Appendix C. Our empirical analysis focuses on three primary dimensions: (1) reasoning performance (Section 5.1), (2) adaptability in sparse-reward environments (Section 5.2), and (3) computational cost (Section 5.3).

### 5.1. Performance and Efficiency in Diverse Tasks

**Overcoming Single-Axis Limitations.** As shown in Table 1, ROSA2 consistently outperforms the single-axis baselines, namely TextGrad (Words-only) and ROSA (Weights-only), across all evaluated model sizes (0.5B to 8B) and domains. This performance gap validates our core hypothesis regarding error attribution. TextGrad is effective at refining prompts but frequently hits a *capability ceiling* on complex tasks where the frozen model simply lacks the intrinsic knowledge to execute the instruction. Conversely, ROSA updates parameters on potentially ambiguous inputs and therefore tends to overfit to noise, leading to the stagnation observed in Figure 4. ROSA2 systematically breaks these bottlenecks because the semantic updates unlock the true potential for parameter adaptation, enabling rapid improvements on challenging tasks. Furthermore, this superiority extends beyond basic test-time adaptation methods. ROSA2 consistently surpasses stronger test-time scaling and self-correction paradigms, including Best-of-16 sampling and

*Table 1.* **Main Results on Standard Reasoning Benchmarks.** We report the accuracy (%) across mathematical (MATH, MATH-500), general (MMLU-R, SuperGPQA), multilingual (MT-AIME24, MT-MATH100), and code generation (HumanEval) tasks. The gains are calculated relative to the Baseline. Best scores are **bolded**, and second-best scores are underlined.

| Model | Method | Mathematical Reasoning | | General Reasoning | | Multilingual Reasoning | | Code Gen. |
| | | MATH | MATH-500 | MMLU-R | SuperGPQA | MT-AIME24 | MT-MATH100 | HumanEval |
| --- | --- | --- | --- | --- | --- | --- | --- | --- |
| Qwen2.5-0.5B -Instruct | Baseline | 23.0 | 24.0 | 9.4 | 3.8 | 2.6 | 15.4 | 31.1 |
| | TextGrad | 31.2 (+8.2) | 29.6 (+5.6) | 12.4 (+3.0) | 3.8 (+0.0) | 2.2 (-0.4) | 18.4 (+3.0) | 36.6 (+5.5) |
| | ROSA | 29.2 (+6.2) | 30.4 (+6.4) | 11.4 (+2.0) | 4.0 (+0.2) | 3.8 (+1.2) | 19.6 (+4.2) | 38.4 (+7.3) |
| | ROSA2 | 40.8 (+17.8) | 39.6 (+15.6) | 18.4 (+9.0) | 6.4 (+2.6) | 4.4 (+1.8) | 25.2 (+9.8) | 44.5 (+13.4) |
| Qwen3-0.6B -Instruct | Baseline | 19.6 | 22.4 | 24.0 | 3.8 | 3.2 | 26.2 | 41.5 |
| | TextGrad | 65.0 (+45.4) | 62.0 (+39.6) | 46.4 (+22.4) | 3.8 (+0.0) | 7.0 (+3.8) | 62.2 (+36.0) | 65.8 (+24.4) |
| | ROSA | 66.2 (+46.6) | 63.0 (+40.6) | 48.8 (+24.8) | 4.0 (+0.2) | 7.2 (+4.0) | 62.0 (+35.8) | 72.0 (+30.5) |
| | ROSA2 | 70.8 (+51.2) | 71.6 (+49.2) | 50.0 (+26.0) | 6.4 (+2.6) | 9.6 (+6.4) | 73.4 (+47.2) | 81.7 (+40.2) |
| Qwen2.5-7B -Base | Baseline | 47.0 | 49.4 | 39.8 | 17.8 | 17.0 | 60.4 | 57.9 |
| | TextGrad | 54.8 (+7.8) | 54.0 (+4.6) | 60.2 (+20.4) | 46.4 (+28.6) | 37.6 (+20.6) | 75.4 (+15.0) | 72.0 (+14.0) |
| | ROSA | 63.4 (+16.4) | 62.4 (+13.0) | 60.2 (+20.4) | 47.8 (+30.0) | 37.0 (+20.0) | 70.4 (+10.0) | 74.4 (+16.5) |
| | ROSA2 | 68.4 (+21.4) | 67.2 (+17.8) | 63.0 (+23.2) | 48.8 (+31.0) | 37.8 (+20.8) | 78.2 (+17.8) | 79.9 (+22.0) |
| Qwen3-8B | Baseline | 50.0 | 42.8 | 57.0 | 24.2 | 29.4 | 75.2 | 78.0 |
| | TextGrad | 63.4 (+13.4) | 62.4 (+19.6) | 70.6 (+13.6) | 40.0 (+15.8) | 40.0 (+10.6) | 81.2 (+6.0) | 82.3 (+4.3) |
| | ROSA | 62.2 (+12.2) | 60.8 (+18.0) | 75.8 (+18.8) | 38.6 (+14.4) | 38.6 (+9.2) | 88.4 (+13.2) | 83.7 (+5.6) |
| | ROSA2 | 80.8 (+30.8) | 80.6 (+37.8) | 84.4 (+27.4) | 52.4 (+28.2) | 44.4 (+15.0) | 93.6 (+18.4) | 88.4 (+10.4) |
| DeepSeek-R1 -Distill-Llama-8B | Baseline | 27.6 | 22.8 | 23.6 | 10.4 | 4.8 | 17.8 | 25.0 |
| | TextGrad | 34.0 (+6.4) | 31.6 (+8.8) | 43.4 (+19.8) | 20.8 (+10.4) | 16.2 (+11.4) | 30.4 (+12.6) | 39.0 (+14.0) |
| | ROSA | 37.8 (+10.2) | 37.6 (+14.8) | 42.8 (+19.2) | 21.4 (+11.0) | 17.2 (+12.4) | 38.6 (+20.8) | 39.3 (+14.3) |
| | ROSA2 | 54.2 (+26.6) | 54.6 (+31.8) | 59.4 (+35.8) | 35.0 (+24.6) | 21.4 (+16.6) | 50.6 (+32.8) | 40.2 (+15.2) |

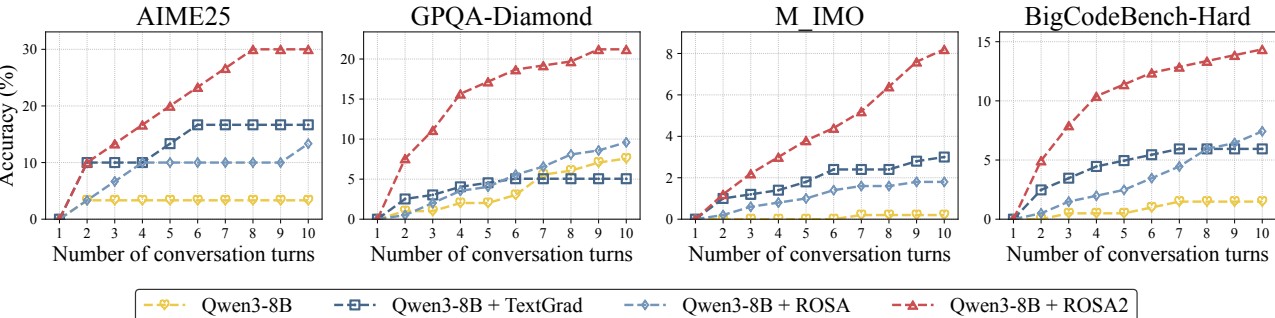

*Figure 4.* **Performance trajectory on challenging benchmarks.** We plot the accuracy on AIME25, GPQA-Diamond, M_IMO, and BigCodeBench-Hard as a function of interaction turns. ROSA2 (red line) demonstrates sustained accuracy improvements, successfully solving complex problems where baselines plateau.

offline fine-tuning, demonstrating that joint co-adaptation breaks the performance ceiling of static scaling laws (Appendix C.7). Similarly, ROSA2 achieves higher interaction efficiency and final accuracy compared to advanced reinforcement learning baselines like SDPO without requiring an expensive prior training stage (Appendix C.8).

**Pre-Conditioning Effect.** To understand the source of our efficiency, we analyze the interaction dynamics detailed in Table 2. ROSA2 achieves the highest *Correction Uplift* (e.g., 81.4% on MATH), confirming that the *Semantic Stream* successfully rectifies initial misunderstandings. More importantly, ROSA2 significantly reduces the *Avg Turn* required to reach a solution (e.g., a 40% reduction compared to ROSA). This reduction provides direct empirical validation for Theorem 4.2. As the interaction progresses,

the continuous semantic refinement actively suppresses the gradient estimation noise, ensuring that the cumulative *Approximation Error* remains significantly lower than that of ROSA. Consequently, this minimization leads to a tighter alignment with the latent user policy $\pi^*_{user}$, which directly translates into the observed higher correction rates and lower turn counts. Crucially, this efficiency is highly robust to optimization randomness. Evaluations across multiple independent random seeds confirm that ROSA2 consistently maintains a exceptionally low variance in both final accuracy and average turns, firmly validating the stability of our co-adaptation loop (Appendix C.6).

**Failure Analysis on Hard Benchmarks.** Figure 4 also reveals the boundary of test-time co-adaptation. On MT-AIME24, the learning curve plateaus when the model en-

*Table 2.* **Analysis of Interaction Dynamics on Qwen3-8B. Correction Uplift** indicates the percentage of eventually solved problems that were corrected after the initial failure. **Avg Turn** denotes the average interaction turns required to solve a problem.

| Dataset | Method | Correction Uplift (↑) | Avg Turn (↓) |
|---|---|---|---|
| **MATH** | Baseline | 70.0% | 7.2 |
| | TextGrad | 75.1% (+5.1%) | 6.0 (-1.2) |
| | ROSA | 77.3% (+7.3%) | 6.3 (-0.9) |
| | ROSA2 | **81.4%** (+11.4%) | **4.4** (-2.8) |
| **MMLU** | Baseline | 50.9% | 6.6 |
| | TextGrad | 59.5% (+8.6%) | 5.2 (-1.4) |
| | ROSA | 60.7% (+9.8%) | 5.0 (-1.6) |
| | ROSA2 | **64.9%** (+14.0%) | **4.1** (-2.5) |
| **MT-AIME24** | Baseline | 66.7% | 9.0 |
| | TextGrad | 74.5% (+7.8%) | 7.9 (-1.1) |
| | ROSA | 73.1% (+6.4%) | 8.2 (-0.8) |
| | ROSA2 | **77.5%** (+10.8%) | **7.7** (-1.3) |

counters cross-lingual mathematical concepts that are poorly represented in the base model. Semantic refinement can clarify the intent and reduce ambiguity, but it cannot synthesize entirely missing multilingual mathematical knowledge from a single interaction trajectory. A similar phenomenon appears on BigCodeBench-Hard, where late-turn stagnation often corresponds to missing knowledge regarding niche APIs or external library behaviors. These failure cases clarify that ROSA2 is most effective at eliciting, aligning, and stabilizing latent capabilities under evolving feedback. It does not replace external knowledge retrieval when the requisite domain knowledge is fundamentally absent.

## 5.2. Adaptability in Sparse-Reward Environments

We evaluate ROSA2 on UI agent tasks (OSWorld (Xie et al., 2024), AndroidWorld (Rawles et al., 2025)) that are characterized by sparse rewards and precise execution demands. Table 3 confirms robust improvements across both SFT and DPO backbones, highlighting the absolute necessity of joint optimization. Single-axis methods universally fail in these environments. TextGrad (Words-only) cannot rectify low-level motor precision errors, while ROSA (Weights-only) struggles to converge given the highly sparse signals.

ROSA2 effectively navigates this dilemma by leveraging **Semantic Pre-conditioning** to bridge the reward gap. Specifically, the *Textual Optimization* module retrospectively analyzes the sequence of unrewarded actions and synthesizes fine-grained corrective instructions that pinpoint specific execution failures. This process effectively densifies the feedback, transforming a vague, delayed failure signal into a detailed supervision signal for the next attempt. Consequently, the *Parameter Optimization* stream utilizes this clarified context as a pre-conditioner to fine-tune the execution policy with exact precision, rather than blindly searching within a sparse reward landscape. This operational synergy, where semantic retrospective feedback

directly guides parametric actuation, is the fundamental reason ROSA2 achieves superior adaptability in complex agentic tasks.

*Table 3.* Adaptability in sparse-reward environments (UI Agents).

| Model | OSWorld | AndroidWorld |
|---|---|---|
| UI-TARS-7B-SFT (Qin et al., 2025) | 13.2 | 27.6 |
| UI-TARS-7B-SFT + TextGrad | 13.7 (+0.5) | 28.3 (+0.7) |
| UI-TARS-7B-SFT + ROSA | 17.8 (+4.6) | 30.9 (+3.3) |
| UI-TARS-7B-SFT + ROSA2 | **23.6** (+10.4) | **35.3** (+7.7) |
| UI-TARS-7B-DPO | 14.8 | 28.9 |
| UI-TARS-7B-DPO + TextGrad | 14.9 (+0.1) | 28.7 (-0.2) |
| UI-TARS-7B-DPO + ROSA | 18.0 (+3.2) | 31.7 (+2.8) |
| UI-TARS-7B-DPO + ROSA2 | **24.4** (+10.6) | **36.6** (+7.7) |

### 5.3. Computational Cost Analysis

Finally, we analyze the practical deployment costs in terms of latency and memory. As shown in Table 4, ROSA2 achieves a remarkable reduction in *Avg Time* per problem. This gain is driven by two synergistic factors. First, the continuous optimization of *Words and Weights* enables the model to resolve problems using significantly more concise *Chain-of-Thought* (CoT) trajectories, drastically cutting the per-turn inference time (intra-turn efficiency). Second, the framework substantially reduces the total conversation turns required for success (inter-turn efficiency). Regarding memory, ROSA2 introduces negligible overhead, peaking at a maximum of +3.1 GB on the MATH dataset. Because the textual and parametric updates consume the same failure trajectory, they can be executed entirely in parallel during the adaptation phase. This parallel preparation ensures that the framework scales efficiently to larger models, such as Qwen3-32B, without disrupting the token-by-token streaming experience for the end user (Appendix D).

*Table 4.* Computational Cost Analysis on Qwen3-8B.

| Dataset | Method | Avg Time (s) (↓) | Peak Memory (GB) (↓) |
|---|---|---|---|
| **MATH** | Baseline | 334.5 | 90.6 |
| | ROSA2 | **297.6** (-36.9) | 93.7 (+3.1) |
| **AIME25** | Baseline | 557.4 | 94.9 |
| | ROSA2 | **467.2** (-90.2) | 95.4 (+0.5) |
| **HumanEval** | Baseline | 543.7 | 94.8 |
| | ROSA2 | **521.3** (-22.4) | 95.2 (+0.4) |
| **BigCodeBench-Hard** | Baseline | 677.9 | 95.2 |
| | ROSA2 | **590.6** (-87.3) | 95.5 (+0.3) |

## 6. Conclusion and Limitations

We introduced ROSA2, a joint optimization framework over context and parameters that resolves the error attribution dilemma. By bypassing the local minima of conditional baselines, ROSA2 achieves state-of-the-art accuracy and interaction efficiency across diverse benchmarks. However, because it strictly aligns latent capabilities rather than retrieving external information, adaptation plateaus when core domain knowledge is absent.

## Impact Statement

This paper presents work whose goal is to advance the field of Machine Learning, specifically within the domain of test-time adaptation for multi-turn interactions. Our framework demonstrates that co-adapting context and parameters can unlock state-of-the-art performance on reasoning and agentic benchmarks. While this work primarily contributes to the technical efficiency and accuracy of LLMs, it also highlights the potential for more capable UI agents. We believe there are no specific negative societal consequences that must be highlighted here, beyond the general considerations associated with the deployment of increasingly capable generative AI models.

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

## A. Related Work

**Adaptation via Context Refinement (Words).** Approaches focusing on the "Words" axis, broadly categorized under Prompt Engineering, aim to optimize the external context $x$ while keeping the model parameters $\theta$ frozen. Yi et al. (2025) review the progression of these methods from manual instruction design to automated strategies that dynamically refine inputs to align with user needs. Recent work has further emphasized the importance of clarifying input intent; for instance, Tang et al. (2025b) investigate the role of ambiguity types in prompting, demonstrating that sharpening semantic clarity can improve generation quality. However, these context-centric optimization methods face a fundamental theoretical ceiling: they cannot induce capabilities that do not exist within the frozen parameters. As argued by Chen et al. (2025) and Lee et al. (2025), semantic refinement alone is insufficient to remedy intrinsic capability deficits. Consequently, such methods often plateau in what Wei et al. (2025b) describe as a "Deficit Trap," where the model understands the task intent but lacks the execution capacity to solve it.

**Adaptation via Parameter Updates (Weights).** Conversely, the paradigm of Test-Time Training (TTT) or Test-Time Policy Adaptation ($T^2PAM$) focuses on the "Weights" axis, allowing for the real-time update of internal parameters $\theta$ during inference. Wei et al. (2025b) introduced ROSA, a method that employs low-rank adaptation (LoRA) to minimize the divergence from a reward-weighted policy, effectively bridging the capability gap observed in frozen models. Similarly, Zuo et al. (2025) proposed Test-Time Reinforcement Learning (TTRL), which treats each interaction turn as a policy optimization step driven by reward signals. While these parameter-centric approaches offer a mechanism to enhance intrinsic model capabilities, they are highly sensitive to the quality of the learning signal. Li et al. (2025a) highlight that performing parameter updates on noisy or ambiguous interaction histories often leads to the learning of spurious correlations. Without the pre-conditioning of a clear context, these methods are prone to gravitating towards an "Overfitting Trap," resulting in performance degradation over extended interaction turns.

**Joint Optimization of Prompts and Parameters.** Recent literature has begun exploring the intersection of prompt and parameter tuning to overcome single-axis limitations. For instance, MetaTuner (Bo et al., 2026) propose a co-optimization framework for large language models. However, such approaches primarily focus on static or training-time joint optimization paradigms. In contrast, ROSA2 is specifically designed for dynamic, multi-turn test-time adaptation. Crucially, ROSA2 does not merely optimize prompts and weights in parallel; it treats the semantic stream as a feedback normalizer that pre-conditions the multi-turn interaction trajectory. This cross-turn coupling ensures that the parametric stream is updated based on clarified intents rather than raw, noisy user feedback, effectively addressing the error attribution dilemma unique to real-time interactions.

## B. Proofs

### B.1. Proof of Theorem 4.1

The proof follows from the closed-form solution of the linearized parameter update in ROSA.

**Step 1: The Residual-Driven Update.** According to Eq. (6) in ROSA (Wei et al., 2025b), the parameter update $\Delta\theta$ is the least-squares solution to fitting the residual between the target distribution $\tilde{\pi}^*$ and the current policy $\pi$:

$$(J^T J)\Delta\theta = J^T R(x), \quad \text{where } R(x) = \tilde{\pi}^*(\cdot|x) - \pi(\cdot|x, \theta_{t-1}) \tag{11}$$

The magnitude of the update is bounded by the magnitude of this residual vector $R(x)$:

$$||\Delta\theta_t(x)||_2 \leq \frac{1}{\sigma_{\min}(J)} ||R(x)||_2 \tag{12}$$

**Step 2: Effect of Semantic Refinement.** The Semantic Stream updates $x_t \rightarrow x_t^*$ to minimize the semantic discrepancy, effectively bringing the current policy's distribution closer to the user's optimal policy $\pi_{user}^*$. Since the target $\tilde{\pi}^*$ is constructed based on $\pi_{user}^*$, reducing the distance to $\pi_{user}^*$ also reduces the distance to $\tilde{\pi}^*$. Therefore, the refined query yields a smaller residual vector:

$$||R(x_t^*)||_2 < ||R(x_t)||_2 \tag{13}$$

**Step 3: Conclusion.** Substituting the reduced residual back into the bound from Step 1, we obtain:

$$||\Delta\theta_t(x_t^*)||_2 \leq C \cdot ||R(x_t^*)||_2 < C \cdot ||R(x_t)||_2 \approx ||\Delta\theta_t(x_t)||_2 \tag{14}$$

Thus, optimizing the query reduces the norm of the required parameter update. $\square$

## B.2. Proof of Theorem 4.2

The proof relies on decomposing the total error into a "theoretical improvement" component and an "approximation error" component. We analyze the change in KL divergence at step $t$ by introducing the theoretical target $\tilde{\pi}_t^*$ as an intermediate point. Using the telescoping sum property, the total error after $T$ turns can be written as:

$$
\begin{aligned}
D_{KL}(\pi_{user}^* || \pi_{\phi_T}) - D_{KL}(\pi_{user}^* || \pi_{\phi_0}) &= \sum_{t=1}^{T} \left( D_{KL}(\pi_{user}^* || \pi_{\phi_t}) - D_{KL}(\pi_{user}^* || \pi_{\phi_{t-1}}) \right) \\
&= \sum_{t=1}^{T} \bigg( \underbrace{D_{KL}(\pi_{user}^* || \tilde{\pi}_t^*) - D_{KL}(\pi_{user}^* || \pi_{\phi_{t-1}})}_{\text{Term A: Ideal Gain}} \\
&\quad + \underbrace{D_{KL}(\pi_{user}^* || \pi_{\phi_t}) - D_{KL}(\pi_{user}^* || \tilde{\pi}_t^*)}_{\text{Term B: Approximation Cost}} \bigg)
\end{aligned}
\tag{15}
$$

**Bounding Term A.** Since our target construction follows reward-weighted regression, we invoke the monotonic improvement argument from ROSA (Wei et al., 2025b). For a positive constant $C$ that absorbs the reward-weighting and normalization terms, the ideal update satisfies:

$$
D_{KL}(\pi_{user}^* || \tilde{\pi}_t^*) - D_{KL}(\pi_{user}^* || \pi_{\phi_{t-1}}) \leq -C \pi_{\theta_{t-1}}(\mathbf{y}_t | \mathbf{x}_t^*)
\tag{16}
$$

The refined instruction $\mathbf{x}_t^*$ strengthens this term because it increases the probability mass assigned to responses aligned with the latent user policy.

**Bounding Term B.** We avoid defining a naive Euclidean norm over the heterogeneous pair $\phi_t = \{\mathbf{x}_t, \theta_t\}$. Instead, we work in the Cartesian product of the continuous token-embedding space and the adapter-parameter space. In a local neighborhood of the current trajectory, the second-order Taylor expansion of the KL divergence is governed by a block Hessian:

$$
H = \begin{bmatrix} H_{xx} & H_{x\theta} \\ H_{\theta x} & H_{\theta\theta} \end{bmatrix}.
\tag{17}
$$

Assume the spectral norms of the blocks are locally bounded as $\|H_{xx}\|_2 \leq L_x$, $\|H_{\theta\theta}\|_2 \leq L_\theta$, and $\|H_{x\theta}\|_2 \leq L_{x\theta}$. Then the approximation cost is bounded by the corresponding quadratic form:

$$
\text{Term B} \leq \frac{L_x}{2} \|\Delta x_t\|_2^2 + \frac{L_\theta}{2} \|\Delta \theta_t\|_2^2 + L_{x\theta} \|\Delta x_t\|_2 \|\Delta \theta_t\|_2.
\tag{18}
$$

Applying Young's inequality to the cross term gives:

$$
L_{x\theta} \|\Delta x_t\|_2 \|\Delta \theta_t\|_2 \leq \frac{L_{x\theta}}{2} \|\Delta x_t\|_2^2 + \frac{L_{x\theta}}{2} \|\Delta \theta_t\|_2^2.
\tag{19}
$$

Therefore:

$$
\text{Term B} \leq \left( \frac{L_x + L_{x\theta}}{2} \right) \|\Delta x_t\|_2^2 + \left( \frac{L_\theta + L_{x\theta}}{2} \right) \|\Delta \theta_t\|_2^2.
\tag{20}
$$

Defining $\bar{L} = \max(L_x + L_{x\theta}, L_\theta + L_{x\theta})$ recovers the decoupled bound:

$$
\text{Term B} \leq \frac{\bar{L}}{2} \left( \|\Delta x_t\|_2^2 + \|\Delta \theta_t\|_2^2 \right).
\tag{21}
$$

Summing these bounds over $t = 1 \ldots T$ yields Theorem 4.2. $\qquad\square$

## B.3. Local Smoothness and Empirical Validation

The bound above requires only local smoothness along the adaptation trajectory, not a global Lipschitz assumption over the entire LLM parameter space. This local condition is natural in ROSA2 because the adapter is initialized at zero, the LoRA rank is small, and the number of interaction turns is bounded by $T_{\max} = 10$. Empirically, both semantic and parametric update magnitudes remain small and decay over turns, supporting the local approximation used in Theorem 4.2.

*Table 5.* Empirical validation of local smoothness on MATH using Qwen3-8B.

| Interaction Turn | Parameter Update | Semantic Update | Policy Shift |
|---|---|---|---|
| Turn 1 | 0.0352 | 0.0018 | 0.142 |
| Turn 5 | 0.0114 | 0.0004 | 0.051 |
| Turn 10 | 0.0021 | $< 0.0001$ | 0.008 |

**Advantage: Locality and Stable Micro-Updates.** Table 5 shows that the parameter update, semantic update, and resulting policy shift all decrease monotonically as the interaction proceeds. This behavior supports the central design of ROSA2: semantic refinement performs a fast macro-level correction early in the dialogue, while the low-rank adapter performs small and stable micro-updates. Therefore, the theory does not rely on an unrealistic global smoothness assumption over the full LLM; it only requires a local neighborhood induced by bounded turns, low-rank updates, and progressively stabilized Words. This also explains why the approximation error in Figure 3 decays instead of accumulating across turns.

# C. Experimental Setup

To rigorously evaluate the efficacy, efficiency, and generalizability of ROSA2, we conducted comprehensive experiments across a wide spectrum of tasks and model architectures. This section details the datasets, models, evaluation metrics, and reward mechanisms employed in our study.

## C.1. Datasets

We assessed ROSA2 on a diverse suite of challenging benchmarks categorized into four distinct domains: Mathematical Reasoning, General Reasoning, Code Generation, and Multilingual Reasoning. Table 6 summarizes the statistics of these datasets.

*Table 6.* Summary of benchmarks used for evaluation. "N/A" denotes datasets primarily used for testing that lack a standard pre-defined training split.

| Domain | Dataset | Train Size | Test Size |
|---|---|---|---|
| Mathematical Reasoning | MATH | 7,500 | 5,000 |
| | AIME25 | N/A | 30 |
| | MATH-500 | N/A | 500 |
| General Reasoning | GPQA-diamond | N/A | 198 |
| | MMLU-Redux | N/A | 3,000 |
| | SuperGPQA | 26,500 | N/A |
| Code Generation | HumanEval | N/A | 164 |
| Multilingual Reasoning | MCLM | N/A | 156 |

**Mathematical Reasoning.** This domain targets complex, multi-step problem-solving. We employed three standard benchmarks:

- MATH (Hendrycks et al., 2021b): A collection of 12,500 challenging high-school level competition problems spanning algebra, geometry, and calculus.

- AIME25 (AIME, 2025): A curated subset of 25 extremely difficult problems from the American Invitational Mathematics Examination, designed to probe advanced reasoning limits.

- MATH-500 (Lightman et al., 2023): A widely recognized evaluation subset of the MATH test set, consisting of 500 problems selected for efficient model assessment.

**General Reasoning.** To evaluate knowledge application across broad topics, we utilized three expert-level QA datasets:

- `GPQA-diamond` (Rein et al., 2024): A high-difficulty set of graduate-level questions written by domain experts; the "diamond" subset ensures the highest quality.

- `MMLU-Redux` (Hendrycks et al., 2021a): A refined version of the Massive Multitask Language Understanding benchmark, covering 57 subjects ranging from elementary math to professional law.

- `SuperGPQA` (Team et al., 2025): An expansion of GPQA containing nearly 5,000 expert-validated questions across 285 graduate-level disciplines.

**Code Generation.** We assessed code synthesis capabilities using `HumanEval` (Chen et al., 2021). This benchmark comprises 164 hand-written programming problems equipped with function signatures, docstrings, and unit tests to verify functional correctness.

**Multilingual Reasoning.** Cross-lingual reasoning was evaluated using `MCLM` (Son et al., 2025), which translates challenging English benchmarks into multiple languages. Our evaluation specifically focuses on the multilingual versions of IMO, AIME, and MATH problems (`M-IMO`, `MT-AIME24`, and `MT-MATH100`).

**Evaluation Protocol.** To simulate real-world deployment, our primary evaluation is conducted on official, held-out test sets. In cases where a dedicated test set is unavailable, or for specific ablation studies, we utilized corresponding training or development sets. Specifically, for `SuperGPQA`, we sampled a portion of the training data for testing purposes; for all other benchmarks, standard test sets were strictly observed.

## C.2. Models

We selected a diverse array of open-source Large Language Models (LLMs) to ensure the robustness of our findings irrespective of model architecture or scale. As detailed in Table 7, our selection includes instruction-tuned variants designed for chat and instruction-following tasks. Note that to mitigate potential data contamination concerns with the `Qwen2.5` series on specific benchmarks, we also validated results using the more recent `Qwen3` and `DeepSeek-R1` models.

*Table 7.* Categorization of language models used in experiments.

| Category | Model | Params | Type |
|---|---|---|---|
| Small-Scale | `Qwen2.5-0.5B-Instruct` | 0.5B | Instruct |
| | `Qwen3-0.6B` | 0.6B | Base |
| Large-Scale | `Qwen2.5-7B-Instruct` | 7B | Instruct |
| | `Qwen3-8B` | 8B | Base |
| Reasoning-Focused | `DeepSeek-R1-Distill-Llama-8B` | 8B | Reasoning |
| | `DeepSeek-R1-Distill-Qwen-7B` | 7B | Reasoning |

**Small-Scale Models.** To evaluate ROSA2 in resource-constrained settings, we selected compact models from the Qwen family: `Qwen2.5-0.5B-Instruct` (Qwen et al., 2025), optimized for instruction following, and `Qwen3-0.6B` (Yang et al., 2025), representing the newer generation with architectural enhancements.

**Large-Scale Models.** We tested scalability using capable base models: `Qwen2.5-7B-Instruct` (Qwen et al., 2025), a standard 7B parameter instruction-tuned model, and its successor `Qwen3-8B` (Yang et al., 2025).

**Reasoning-Focused Models.** We specifically included the DeepSeek-R1 series (DeepSeek-AI et al., 2025), which are optimized via reinforcement learning for complex reasoning. We utilized distilled variants based on both Llama (`DeepSeek-R1-Distill-Llama-8B`) and Qwen (`DeepSeek-R1-Distill-Qwen-7B`) architectures to allow for controlled architectural comparisons.

## C.3. Evaluation Metrics

Our evaluation framework focuses on two critical dimensions: downstream task performance and computational efficiency.

**Performance Metrics.**

- **Accuracy:** Defined as the proportion of unique problems correctly solved within a maximum of $K$ conversational turns. Let $\mathcal{P}$ be the set of problems and $S_i \in \{0, 1\}$ be an indicator variable where $S_i = 1$ if problem $i$ is solved at any turn $t \leq K$. Accuracy is calculated as:

$$\text{Accuracy} = \frac{\sum_{i \in \mathcal{P}} S_i}{|\mathcal{P}|} \tag{22}$$

- **Correction Uplift:** This metric quantifies the model's capacity to self-correct. It represents the percentage of problems initially answered incorrectly that were subsequently solved in later turns. Let $\mathcal{P}_{\text{fail}} \subset \mathcal{P}$ denote problems failed at turn $t = 1$. The metric is defined as:

$$\text{Correction Uplift} = \frac{\sum_{i \in \mathcal{P}_{\text{fail}}} S_i}{|\mathcal{P}_{\text{fail}}|} \times 100\% \tag{23}$$

**Efficiency Metrics.** To measure computational overhead, we track:

- **Avg Time :** The average time solve per problem.

- **Peak GPU Memory:** The maximum VRAM usage observed during the inference and update process.

### C.4. Reward Models

We employed two distinct reward mechanisms to simulate varying feedback granularities found in real-world applications.

**Rule-Based Reward Model (Sparse Feedback).** This model simulates scenarios with definitive, binary judgments. It programmatically extracts the final answer (e.g., from a \boxed{} environment) and matches it against the ground truth. A reward of $+1.0$ is assigned for an exact match, and $-1.0$ otherwise. The core implementation logic is provided below.

---

**Core logic for the rule-based reward model**

```python
class MathVerifyRewardModel:
    def __init__(self, ground_truth_answer: str):
        self.ground_truth_answer = ground_truth_answer

    def get_reward(self, response_text: str) -> float:
        # Returns +1.0 for match, -1.0 otherwise
        return 1.0 if compute_score(response_text,
            self.ground_truth_answer) == 1.0 else -1.0

def compute_score(solution_str, ground_truth) -> float:
    retval = 0.0
    try:
        string_in_last_boxed = last_boxed_only_string(solution_str)
        if string_in_last_boxed is not None:
            answer = remove_boxed(string_in_last_boxed)
            if is_equiv(answer, ground_truth):
                retval = 1.0
    except Exception:
        pass
    return retval
```

---

### C.5. Reproducibility, Hyperparameters, and Fairness Protocol

All multi-turn methods use the same base checkpoint, the same initial prompt, the same interaction budget, and exactly one response per turn. The only exception is Best-of-16, which is explicitly labeled as a static test-time sampling baseline and receives 16 independent generations in a single turn. The TextGrad critic is the current base model itself, not an external oracle.

*Table 8.* Comprehensive hyperparameter setup for ROSA2 and core baselines.

| Hyperparameter | Value / Configuration |
|---|---|
| Interaction Limit ($T_{\max}$) | 10 turns |
| Responses per Turn | $N = 1$ for all multi-turn methods |
| LoRA Rank ($r$) | 1 |
| LoRA Alpha ($\alpha$) | 8 |
| Learning Rate (Param Optimization) | $1 \times 10^{-4}$ |
| KL Penalty Coefficient ($\beta$) | 1.0 |
| TextGrad Critic Model | Current base model |
| TextGrad Optimization Steps / Turn | 1 step |
| Best-of-N Baseline | $N = 16$ independent generations |
| SDPO Train-time Sampling | $N = 8$ responses per prompt |

**Advantage: Strictly Self-Contained Adaptation.** The hyperparameter setup highlights that ROSA2 does not rely on a stronger external critic or a hidden inference budget. The same base model is used for generation, semantic refinement, and parametric adaptation, which isolates the gain to the co-adaptation mechanism itself. The extremely small LoRA rank ($r = 1$) further emphasizes that ROSA2 does not obtain its gains by introducing a large trainable module; instead, it uses Words to reduce the difficulty of the required Weight update.

**Multi-turn Baseline.** The standard inference baseline is also a multi-turn interaction baseline. At each turn, if the model fails, we append the uniform feedback prompt: "Wrong answer, please rethink and try another way of thinking!" This ensures that the baseline receives the same interaction budget as the adaptive methods, but without updating either Words or Weights.

This baseline isolates the value of adaptation from the value of merely having more turns. Its poor performance shows that repeated prompting with generic feedback cannot reliably overcome the Deficit Trap: the model may see the same failure signal many times, but without a refined semantic trajectory or a parameter update, it remains constrained by the frozen policy.

**Zero-shot Protocol and Official Benchmark Differences.** The Qwen3-8B baseline on MATH is lower than the official report because our evaluation uses a simple zero-shot CoT template to simulate realistic users who begin with a naive query and refine it through interaction. Official reports often use highly engineered few-shot prompts or majority voting. All methods in our comparison use the same zero-shot template, so relative improvements are directly comparable.

This setup stresses the practical advantage of ROSA2: instead of assuming that the user already provides an expert prompt, ROSA2 turns a naive starting point into an increasingly clarified interaction. The improvement therefore reflects test-time elicitation and alignment of latent capabilities, rather than additional prompt engineering performed before deployment.

**Negative Feedback Updates.** ROSA2 can update parameters from a single rejected trajectory. The target distribution down-weights the failed response through the reward-weighted term $\exp(r/\beta)$, thereby reducing the probability of repeating that specific incorrect trajectory. The Words stream provides the complementary positive direction by converting the failure history into a refined instruction for the next turn.

This mechanism is important in realistic multi-turn dialogue, where users often provide only a rejection or a short complaint rather than a correct reference answer. The Parametric Stream supplies a negative push away from the observed failure, while the Words Stream supplies a positive pull toward a clearer next attempt. Their combination allows ROSA2 to learn from errors without requiring multiple positive samples per turn.

### C.6. Robustness Across Random Seeds

We rerun the core Qwen3-8B/MATH interaction dynamics experiment across five independent random seeds. The previously drafted value $68.4 \pm 0.3$ for ROSA2 final accuracy was a typo; the correct value is $80.8 \pm 0.3$.

*Table 9.* Robustness across five random seeds on Qwen3-8B/MATH. Final Acc. is solved-within-$T_{\max}$ accuracy, while Correction Uplift is computed only over problems that fail at Turn 0.

| Method | Final Acc. | Correction Uplift | Avg Turn |
|--------|-----------|-------------------|----------|
| Baseline | $50.0 \pm 5.3$ | $70.0 \pm 3.9$ | $7.2 \pm 1.2$ |
| TextGrad | $63.4 \pm 1.7$ | $75.1 \pm 0.6$ | $6.0 \pm 0.2$ |
| ROSA | $62.2 \pm 0.9$ | $77.3 \pm 0.8$ | $6.3 \pm 0.3$ |
| ROSA2 | $\mathbf{80.8 \pm 0.3}$ | $\mathbf{81.4 \pm 0.5}$ | $\mathbf{4.4 \pm 0.1}$ |

**Advantage: Robustness to Optimization Randomness.** The standard deviations in Table 9 are small for both final accuracy and average turns, especially for ROSA2. This indicates that the gain is not caused by a lucky prompt trajectory or a favorable adapter initialization. Instead, the co-adaptation loop consistently reduces the number of interaction turns and improves correction uplift across random seeds. The low variance of the average turn metric is particularly important for deployment, because it means the user experience is predictably shorter rather than occasionally improved.

### C.7. Stronger Baselines and Semantic-Stream Ablations

To further stress-test the practical advantage of ROSA2, we compare against static test-time scaling, offline fine-tuning, and stronger prompt-only self-correction baselines. The TextGrad row in Table 10 follows the prompt protocol used to align with Self-Critique/Reflexion; therefore, it is not directly identical to the TextGrad prompt used in the main benchmark table. The row ROSA2 (ROSA + Self-Critique) is a semantic-stream ablation that replaces the default TextGrad semantic optimizer with Self-Critique-style refinement.

*Table 10.* Comparison with stronger optimization baselines on Qwen3-8B/MATH.

| Method | Paradigm | Avg. Time to Solve | Accuracy |
|--------|----------|--------------------|----------|
| Zero-Shot (Base) | Static | 1.0x | 50.0% |
| SFT Before Test | Offline Training | ~1.2x | 64.4% |
| Best-of-16 (Voting) | Test-Time Sampling | ~16.0x | 67.8% |
| Self-Critique / Reflexion | LLM Self-Correction | ~2.1x | 70.8% |
| TextGrad | LLM Self-Correction | ~2.2x | 71.4% |
| ROSA2 (ROSA + Self-Critique) | Joint Optimization | ~4.1x | **81.6%** |
| ROSA2 (ROSA + TextGrad) | Joint Optimization | ~4.1x | 80.8% |

**Advantage: Breaking the Prompt-Only Ceiling.** Table 10 shows that static test-time scaling (Best-of-16), offline SFT, and prompt-only self-correction all improve over the zero-shot baseline, but they remain below ROSA2. This pattern supports the Deficit Trap hypothesis: when the correct reasoning trajectory is outside the frozen model's active distribution, sampling more outputs or rewriting the prompt alone still hits a ceiling. By contrast, ROSA2 modifies both the semantic trajectory and the local policy, allowing it to move beyond the prompt-only ceiling.

**Advantage: Semantic-Stream Modularity.** The ROSA2 (ROSA + Self-Critique) row shows that the benefit is not tied to a single prompt optimizer. Replacing TextGrad with a Self-Critique-style semantic stream still produces strong performance, demonstrating that the key ingredient is the Words–Weights co-adaptation pattern: semantic refinement prepares a cleaner trajectory, and parametric adaptation internalizes the corrected direction. This modularity suggests that stronger future semantic optimizers can be plugged into ROSA2 without changing the parametric update principle.

### C.8. SDPO Comparison

We also compare against SDPO under a strictly aligned multi-turn setting with $N = 1$ response per test-time turn. We evaluate two SDPO variants: pure test-time adaptation from the base model, and a train+test setting where train-time SDPO is first performed on the MATH training set.

The pure test-time SDPO variant struggles in the strict $N = 1$ regime, while train-time SDPO improves stability at the cost of an additional offline RL stage. In contrast, ROSA2 reaches stronger final accuracy without a prior task-specific training

*Table 11.* Test-time adaptation comparison with SDPO on MATH using Qwen3-8B. Turn-0 Acc. is the correctness of the first response only; Final Multi-turn Acc. is the cumulative solved-within-10-turn accuracy.

| Method | Prior Training | Turn-0 Acc. | Final Acc. | Avg Turns |
|---|---|---|---|---|
| Baseline (Multi-turn Prompting) | None | 15.8% | 50.0% | 7.2 |
| SDPO (Test-time only) | None | 15.8% | 53.6% | 6.7 |
| ROSA2 | None | 15.8% | **80.8%** | **4.4** |
| SDPO (Train + Test) | Train-time RL | 34.0% | 69.0% | 5.9 |

stage.

**Advantage: Train-Free Test-Time Alignment.** The SDPO comparison highlights a different strength from Table 10. SDPO becomes substantially stronger only after reshaping the model with train-time RL on MATH, which changes the initial policy before test-time adaptation begins. ROSA2 starts from the same base model as the multi-turn baseline and still outperforms the train+test SDPO pipeline. This shows that semantic pre-conditioning can stabilize single-trajectory test-time updates without requiring an expensive preparatory RL stage.

**Advantage: Higher Interaction Efficiency.** Even when SDPO improves final accuracy, it requires more turns than ROSA2. The reduction from 5.9 to 4.4 average turns against the trained SDPO pipeline means that ROSA2 not only solves more problems but also reaches acceptance faster. This directly supports our objective of streamlining multi-turn interactions rather than merely improving final solved rate.

## D. Computational Decomposition and Scalability

Table 12 decomposes the per-turn cost on a single H20 GPU with vLLM. ROSA2 preserves token-level streaming because adaptation occurs between turns. The Words-precondition-Weights relation is realized through the interaction trajectory: the current failed trajectory is generated under previously refined Words. Therefore, after a rejection, the semantic update for the next prompt and the parametric update for the next adapter can be executed in parallel, both preparing the next turn.

*Table 12.* Per-turn computational decomposition on one H20 GPU with vLLM.

| Metric per Turn | Qwen3-8B | Qwen3-32B |
|---|---|---|
| Baseline Peak Memory (Generation) | 90.6 GB | 93.1 GB |
| Peak Semantic Update Memory (Words) | 91.2 GB | 93.2 GB |
| Peak Parametric Update Memory (Weights) | 93.6 GB | 94.6 GB |
| ROSA2 Parallel Peak Memory | **93.6 GB (+3.0 GB)** | **94.6 GB (+1.5 GB)** |
| Baseline Avg. Generation Time | 46.4s | 67.2s |
| ROSA2 Avg. Generation Time | 37.6s | 49.9s |
| ROSA2 Avg. Semantic Update Time (Words) | 30.1s | 40.6s |
| ROSA2 Avg. Parametric Update Time (Weights) | 29.6s | 41.8s |
| ROSA2 Parallel Total Turn Time | **67.2s (+20.8s)** | **90.5s (+23.3s)** |

As the conversation proceeds, the context becomes longer and each subsequent generation becomes more expensive. ROSA2 reduces the number of required turns and tends to produce shorter corrected trajectories, which lowers the total generation cost. Although each rejected turn introduces an adaptation phase, the semantic and parametric updates for the next turn can be executed in parallel because both consume the current failed trajectory. This explains why ROSA2 can have a higher per-turn cost but a lower average time per solved problem.

**Advantage: Streaming-Preserving Adaptation.** The decomposition shows that ROSA2 does not interrupt token-by-token decoding. The model first streams a normal response; only after the user rejects the response does the system enter the adaptation phase. Once $x_{t+1}^*$ and $\theta_{t+1}$ are prepared, the next response is again generated with standard autoregressive streaming. This is important for user experience because the adaptation cost appears as a bounded between-turn thinking phase rather than slower token emission.

**Advantage: Parallel Preparation for the Next Turn.** Although Words pre-condition Weights causally across turns, the two updates after a rejection consume the same current failure trajectory and can therefore be run in parallel. This is why the parallel total turn time is much smaller than the sum of generation, semantic update, and parametric update. The Qwen3-32B results also show that the memory overhead remains modest relative to the already allocated generation memory, suggesting that the approach scales beyond the 8B setting evaluated in the main tables.

## E. Open-Ended Feedback and Words-Stream Analysis

To verify that ROSA2 is not restricted to exact-match rewards, we additionally evaluate open-ended dialogue alignment on MT-Bench using GPT-5.2 as an LLM judge to simulate user acceptance. The judge is used only for evaluation and user simulation in this open-ended study; it is not used as the TextGrad critic in the main experiments.

*Table 13.* Open-ended dialogue alignment on MT-Bench with GPT-5.2-as-a-Judge.

| Method | Avg Turn to Acceptance | Win Rate vs. Base |
|---|---|---|
| One-Turn | N/A | 50.0% |
| Standard Prompting | 4.8 | 62.3% |
| TextGrad Only | 3.5 | 71.5% |
| ROSA | 3.8 | 75.5% |
| ROSA2 | **2.1** | **85.4%** |

**Advantage: Beyond Verifiable Binary Rewards.** Table 13 demonstrates that ROSA2 can operate when no exact ground-truth answer exists. The improvement in both win rate and average turns indicates that the Words stream can translate subjective critiques into useful adaptation targets. Compared with TextGrad-only and ROSA-only variants, the joint method is better at both understanding the user's evolving preference and internalizing it into the policy for later turns.

**Qualitative Words-Stream Example.** The Words stream converts raw feedback into an actionable instruction before the next response. For example, if the user feedback is "This is not what I meant; make it more practical," the Semantic Stream refines it into: "The user rejects the abstract explanation and wants a concrete, implementation-oriented answer with step-by-step operational details and fewer high-level claims." This refined instruction preserves the user's intent while removing ambiguity, allowing the next response and the subsequent parameter update to target the clarified trajectory.

As a second example, consider the noisy feedback "No, the tone is off and the answer still feels wrong." A direct parameter update on this sentence would be ambiguous: the user may object to style, factual content, or both. The Words stream rewrites it into a structured instruction such as: "Revise the answer to be more concise and neutral in tone; re-check the factual claim before giving the final response." This shows the denoising role of Words: it separates preference correction from factual correction before the next model response is generated. Consequently, the Weights stream is not asked to fit an emotionally phrased complaint directly; it is updated on a clearer trajectory produced by the refined instruction.

**Advantage: Protection Against Noisy Feedback.** These qualitative examples illustrate why Words are necessary even when a binary accept/reject signal is available. A rejection alone tells the system that the previous response failed, but it does not identify the reason. The Semantic Stream transforms that weak signal into a structured correction, preventing the Parametric Stream from overfitting to noisy or underspecified feedback. This is the mechanism by which ROSA2 extends from verifiable tasks to realistic multi-turn conversations.

