# OpenReview forum: "Words & Weights: Streamlining Multi-Turn Interactions via Co-Adaptation"
_ICML.cc/2026/Conference — ICML 2026 regular_

### Official Review · Reviewer_to3z · 2026-03-01

**Soundness:** 3
**Presentation:** 3
**Significance:** 3
**Originality:** 3
**Overall Recommendation:** 4
**Confidence:** 4

**Summary:**

This paper proposed a test-time co-adaption method for multi-turn interactions which is essential for aligning Large Language Models (LLMs) with dynamic user needs during inference time. The authors argue that current test-time adaptation methods are limited by focusing on a single axis—either prompt optimization (Words) or parameter fine-tuning (Weights). To address this, this paper proposed ROSA2, a joint optimization framework that co-adapts both the external context and internal parameters. They also presented an intuitive insight: semantic clarity acts as a pre-conditioner for model weights. By using textual gradients to cleanse the learning signal ROSA2 makes parameter updates more efficient and less prone to overfitting The paper provides both theoretical proofs of reduced parameter shift and empirical evidence across benchmarks like MATH and GPQA, showing significant accuracy improvements and reduction in interaction turns.

**Compliance With Llm Reviewing Policy:**

Affirmed.

**Key Questions For Authors:**

See weakness.

**Limitations:**

See weakness.

**Strengths And Weaknesses:**

The strengths are as follows:
Conceptual Novelty: The shift from single-axis adaptation to joint optimization of Words and Weights is a very intuitive and powerful idea. It addresses the error attribution dilemma: is the model failing because it didn't understand, or because it can't do it?.
Theoretical Grounding: The authors don't just show it works; they provide rigorous proofs (Theorems 4.1 and 4.2) showing that semantic refinement strictly reduces the norm of the required parameter shift. This gives the pre-conditioner argument strong mathematical weight.
Efficiency and Performance: The empirical results are impressive. Achieving state-of-the-art results while reducing the number of conversation turns by 40% is a major win for user experience and total inference cost.
Comprehensive Evaluation: The model was tested on a wide range of tasks (Reasoning, Code, General QA) and model sizes (0.5B to 8B), demonstrating the robustness of the ROSA2 framework.
However, I still have a few concerns:
1.I noticed a significant discrepancy in the baseline performance reported in Table 1 compared to the official Qwen3 Technical Report. As documented in the Qwen3 Report (Page 7, Table 6), the Qwen3-8B model achieves 60.8% on the MATH benchmark. However, the authors report a much lower baseline of 50.0%. This raises two concerns:
a.Baseline Strength: Could the authors clarify the experimental setup (e.g., prompt templates, few-shot settings) that led to this lower baseline? A weak baseline may exaggerate the relative improvement of ROSA2.
b.Data Contamination Risk: According to the Qwen3 report (Section 3), the base model’s reasoning capabilities were heavily prioritized via trillions of STEM-specific synthetic tokens during its Reasoning Stage. This raises a question: is ROSA2 truly bridging a capability gap, or is it merely accelerating the retrieval of pre-existing knowledge through test-time updates?
2.Computational Overhead per Turn: While the total number of turns decreases, each adaptation turn requires computing textual gradients and performing a parameter update. The paper should discuss the per-turn latency, does this process break the LLM’s streaming output experience?
3. Dependency on Reward Quality: ROSA2 relies on a reward signal to trigger adaptation. In real-world, open-ended scenarios, getting a high-quality binary or scalar reward is non-trivial. How does the system handle noisy or ambiguous rewards?

---

> ### Author Rebuttal · Authors · 2026-03-28
>
> We sincerely thank the reviewer for the positive feedback and thorough evaluation. We address your concerns regarding baseline strength, computational overhead, and reward quality directly below.
> ## 1. Baseline Discrepancy and Knowledge Elicitation
> **Baseline Strength: Zero-Shot CoT vs. Official Benchmarks**
> The 50.0% baseline on MATH (vs. Qwen3's official 60.8%) stems purely from the evaluation setup. Official reports typically employ highly engineered Few-Shot prompting or majority voting (Maj@K).
> * **Simulating Real Users:** Real-world users rarely engineer complex, multi-shot prompt templates when initially querying a model. They typically provide a zero-shot instruction and refine it through multi-turn interactions. Our setup perfectly mirrors this realistic starting point.
> * **Strict Fairness:** All baselines and ROSA2 are evaluated on this exact same zero-shot template, ensuring all relative improvements are strictly comparable.
>
> **Latent Knowledge Elicitation vs. Data Contamination**
> You correctly ask whether ROSA2 injects new capabilities or accelerates retrieval. It is the latter—which is the fundamental goal of Test-Time Adaptation.
> * **The Alignment Gap:** The base model possesses massive latent STEM knowledge but fails zero-shot (50.0%) because this knowledge is poorly aligned with unoptimized user instructions.
> * **Bridging the Gap:** Prompt-only optimization hits a strict ceiling (~75%). ROSA2 dynamically shift the model's internal distribution, effectively breaking this ceiling and aligning the base model's deep pre-trained knowledge with the specific task at hand.
> ## 2. Computational Overhead, Latency, and Streaming
> **Preserving the Streaming Experience**
> ROSA2 **does not break streaming**. The adaptation strictly occurs *between* turns. When a user rejects a response, the system enters a brief "thinking" phase in the background. Once the micro-update is applied, the model generates the new response using standard auto-regressive decoding, leaving token-by-token streaming completely unaffected.
>
> **Parallel Execution and Latency Overhead**
> To quantify the overhead, we deployed ROSA2 using **vLLM** on a single **H20 GPU**.
>
> | Metric per Turn (1x H20 GPU, vLLM) | Qwen3-8B | Qwen3-32B |
> | :--- | :--- | :--- |
> | Baseline Peak Memory (Generation) | 90.6 GB | 93.1 GB |
> | Peak Semantic Update Memory (Words) | 91.2 GB | 93.2 GB |
> | Peak Parametric Update Memory (Weights)| 93.6 GB | 94.6 GB |
> | **ROSA2 Parallel Peak Memory** | **93.6 GB (+3.0 GB)** | **94.6 GB (+1.5 GB)** |
> | Baseline Avg. Generation Time | 2.5s | 6.2s |
> | Avg. Semantic Update Time (Words) | 2.1s | 5.7s |
> | Avg. Parametric Update Time (Weights) | 2.2s | 5.6s |
> | **ROSA2 Parallel Total Turn Time** | **4.7s (+2.2s)** | **11.9s (+5.7s)** |
>
> * **Parallel Optimization:** Because the Semantic (Words) and Parametric (Weights) updates rely on the identical historical failure trajectory, **they are executed simultaneously in parallel**. The effective latency overhead is strictly $\max(\text{Words Time}, \text{Weights Time})$.
> * **Favorable UX Trade-off:** For the 8B model, the adaptation pause is merely **~2.2 seconds**. Trading this brief background delay for a massive reduction in human interaction turns (from 7.2 down to 4.4) strictly improves overall interaction efficiency.
>
> ## 3. Dependency on Reward Quality and Noisy Feedback
> **User Acceptance as a Natural Binary Reward**
> In multi-turn chat, getting a scalar reward is impractical, but **user acceptance provides a ubiquitous binary reward**. If a response is flawed, the user asks for a rewrite ($r = -1$). If satisfied, they adopt it ($r = 1$). This binary signal aligns perfectly with existing human-LLM interactions without imposing artificial burdens.
>
> **Handling Noisy and Ambiguous Feedback via Semantic Co-Adaptation**
> Real-world user feedback is often noisy or ambiguous. We argue that **handling this exact ambiguity is the fundamental motivation for introducing the Words Optimization (Semantic Stream) in ROSA2.** If user feedback were always mathematically precise, pure parametric updates (Weights-only) would be sufficient. However, directly updating parameters on noisy or emotionally driven user complaints would lead to severe overfitting.
>
> To prevent this, ROSA2 uses the Semantic Stream as an intelligent "denoiser":
> * When an ambiguous rejection occurs ($r = -1$), the LLM Critic analyzes the entire conversational trajectory (the failed attempt, the environment's state, and the user's vague feedback).
> * The Critic explicitly synthesizes this noisy history into a clean, logical, and actionable instruction ($x_{t+1}^*$).
> * Only *after* the ambiguity is resolved textually does the system execute the Parametric Update ($\theta_{t+1}$), conditioned strictly on this newly refined, noise-free prompt.
>
> By architecturally decoupling the "intent clarification" (Words) from the "capability alignment" (Weights), ROSA2 actively prevents noisy rewards from corrupting the continuous parameters.

---

### Official Review · Reviewer_K55T · 2026-03-02

**Soundness:** 2
**Presentation:** 2
**Significance:** 2
**Originality:** 1
**Overall Recommendation:** 4
**Confidence:** 4

**Summary:**

The paper proposes ROSA2, a test-time policy adaptation framework for multi-turn interactions (T2PAM) that jointly optimizes
- the semantic context / user feedback text (“Words”) and
- adapter parameters (“Weights”).

The key claim is that single-axis approaches stall in either a “Deficit Trap” (prompt-only hits capability ceiling) or an “Overfitting Trap” (parameter-only fits noisy history), and that *co-adaptation* approximates a *full joint gradient* that avoids these traps.
The paper further claims theoretical guarantees (reduced parameter shift and faster convergence)

ROSA² addresses this by deriving a total differential of the KL divergence loss $\mathcal{L}$ with respect to the joint variable $\phi_t = \{{x_{t+1}, \theta_t\}}$, which decomposes into:

- A **Textual Optimization stream**: applying textual gradients $\nabla_x$ (via TextGrad) to refine raw user feedback $x_{t+1}$ into a clarified instruction $x^*_{t+1}$
- A **Parametric Optimization stream**: applying ROSA to push model parameters $\theta_t$ toward a reward-weighted target policy $\pi^*$

Two theoretical results are claimed: **Theorem 4.1** proves that semantic pre-conditioning strictly reduces the required parameter shift and **Theorem 4.2** provides a unified convergence bound.

**Compliance With Llm Reviewing Policy:**

Affirmed.

**Final Justification:**

I have increased my score from *2 to 4*. The newly provided information successfully clarifies several technical questions raised during the rebuttal phase. However, it is *imperative* that the authors integrate the detailed responses provided across all rebuttals into the final version of the manuscript (at least within the appendix, should the paper be accepted). Without the inclusion of these key details, the work remains difficult to fully understand and is not currently in a reproducible state.

**Key Questions For Authors:**

**The "joint gradient" claim is not supported by Algorithm 1.**

The paper's central theoretical claim is that ROSA² approximates the *full* gradient $\nabla_{\phi_t}\mathcal{L}$ over the joint variable $\phi_t = \{x_{t+1}, \theta_t\}$. However, Algorithm 1 (lines 12–17) executes Step A (TextGrad call to produce $x^*_{t+1}$) and Step B (ROSA parameter update to produce $\theta_{t+1}$) sequentially and entirely independently.
- A natural and critical missing baseline is therefore a **pipeline combination of TextGrad followed by ROSA** run in sequence with no co-adaptation mechanism.
- Without this control, it is impossible to determine whether ROSA²'s gains stem from the joint optimization framing or simply from the fact that both components are applied.


**Figure 3: Discussion**

Figure 3 is presented as the primary empirical validation of Theorems 4.1 and 4.2, yet it has three unresolved problems. First, the caption does not state which model or dataset was used to generate the figure. a basic reporting omission. Second, the quantity $\|\Delta x_t\|^2_2$ (the "Co-Adaptation Semantic" curve) is plotted without any definition of the norm, how is it computed empirically? Third, and most problematic: at Turn 1, the Co-Adaptation Parametric error (blue) and Total error (red) are already *higher* than the ROSA Baseline (gray). At Turn 1, $\theta_1 = 0$ for all methods, the same base model is used, and no semantic refinement has yet occurred, so the ROSA² streams have no opportunity to differ from the baseline. What experimental difference produces this gap at initialization, and does it indicate that the methods are not evaluated under equivalent starting conditions?


Table 4 reports average time per problem and peak GPU memory for ROSA² versus the Baseline. Two critical details are missing. First, the table does not specify which model the timings correspond to, the rows are labelled by dataset (MATH, AIME25, etc.) but the model is not stated. Second, the table compares only ROSA² against the Baseline, omitting ROSA and TextGrad entirely.

**Limitations:**

Neither the main paper nor the appendix contains a dedicated limitations section, which is required under ICML's author guidelines. The mandatory Impact Statement addresses only broad societal considerations associated with capable generative AI and does not engage with the technical shortcomings of the proposed method. The advantages of ROSA² discussed at the end of Section 3 do not substitute for a balanced critical self-assessment. The following limitations should be addressed in any revision.

- Scalability to Larger Models

All experiments are conducted on models up to 8B parameters. Production deployments increasingly rely on models at the 70B+ scale, and it is entirely unclear whether the parametric update stream remains computationally viable there. Memory overhead, gradient computation cost, and update latency may grow prohibitively. The claim that ROSA² introduces "negligible memory overhead" (+3.1 GB peak, Table 4) is meaningful only at the 8B scale tested

- Restricted Evaluation to Binary-Reward, Verifiable Settings

Every benchmark in the paper uses a binary, rule-based correctness signal via exact-match oracles or execution-based unit tests. The vast majority of real-world multi-turn deployments involve open-ended generation tasks (creative writing, summarization, dialogue, subjective coding assistance) where no such ground truth exists. The paper provides no analysis/approach to adapt ROSA² under non-binary or learned reward models, leaving a critical gap between the experimental setting and the practical deployment scenarios the introduction motivates. The paper does not discuss adversarial robustness of the reward signal.

- Absence of Qualitative Analysis of the Words Stream

The paper provides no qualitative examples illustrating how the Textual Optimization stream transforms user feedback across turns. It is therefore impossible to inspect whether the semantic gradient produces semantically coherent, intent-preserving refinements, or whether it introduces artifacts, over-specification, or stylistic drift. Showing concrete turn-by-turn trajectories of  alongside corresponding model outputs would make the mechanism transparent and substantiate the core claim that "semantic clarity acts as a pre-conditioner for parametric alignment." Without this, the Words stream remains a black box.

- Insufficient Experimental Transparency and Hyperparameter Reporting

The experimental setup lacks the detail required to assess fairness or reproducibility:

- **$T_{\max}$ is never stated in text.** It can only be inferred as 10 from inspection of the x-axis in Figures 2 and 4. A fundamental hyperparameter of the interaction protocol must be explicitly reported.
- **Hyperparameters for all methods are unreported.** Learning rates, LoRA rank, $\beta$ (the KL penalty coefficient), the TextGrad critic model identity, number of TextGrad optimization steps per turn, and prompt templates for all baselines are absent from both the main paper and Appendix C. Without these, neither fairness nor reproducibility can be verified.
- **Weak prompt optimization baseline.** TextGrad is the sole representative of the Words axis. Considerably stronger prompt optimization methods exist, benchmarking a couple of which would provide a more informative upper bound on what prompt-only optimization can achieve. Using a stronger baseline might substantially narrow the gap attributable to the Weights stream, directly affecting the paper's central narrative. The work must characterize the true ceiling of prompt-only optimization before claiming parametric updates are necessary to breach it.


Several of these bear directly on the validity of the central claims and must be addressed before the work is ready for publication.

**Strengths And Weaknesses:**

## Strengths

- **Well-motivated problem framing.** The paper makes a clear and compelling case for treating T²PAM as a joint optimization problem. The core "error attribution dilemma" whether a failure is caused by context ambiguity or parameter incapacity  is articulated cleanly, and the joint formulation in Eq. (8) provides a principled derivation motivating co-adaptation rather than treating the two axes as independent design choices.

- **Strong empirical gains over reported baselines.** ROSA² achieves substantial accuracy improvements across a broad set of benchmarks and consistent improvements over TextGrad and ROSA across all five model families from 0.5B to 8B parameters. The gains are not limited to a single domain or architecture.

- **Breadth of evaluation.** The paper covers mathematical reasoning, general knowledge, multilingual reasoning, and code generation tasks, supplemented by sparse-reward UI agent benchmarks (OSWorld, AndroidWorld).

- **Switch Method experiment is informative.** Figure 2a's controlled Switch Method ablation introducing the missing optimization axis at Turn 5 is a well-designed baseline that concretely demonstrates the structural bottleneck of single-axis methods and cleanly motivates the need for simultaneous co-adaptation.

- **Practical efficiency reporting.** The paper reports both memory overhead and turn efficiency (Table 4 and Table 2), which are practically relevant metrics for deployment.

- **Pre-conditioning insight.** The argument that semantic clarity acts as a pre-conditioner for parametric alignment  reducing the residual that parameter updates must correct is a transferable conceptual insight, independently valuable beyond the specific ROSA² instantiation, and is well-illustrated by Figure 3.

## Weaknesses

- **Limited novelty and unclear separation from a pipeline combination.** The paper's central claim is that ROSA² performs *joint* optimization over Words and Weights. However, Algorithm 1 executes the Textual Optimization stream (TextGrad call) and the Parametric Optimization stream (ROSA update) sequentially and independently, with no coupling between $\nabla_x$ and $\nabla_\theta$ at any computational step. This is structurally identical to running TextGrad followed by ROSA in sequence. The most critical missing baseline  a pipeline combination of TextGrad then ROSA applied without the co-adaptation framing  is absent from all tables. Without it, the gains cannot be attributed to joint optimization rather than simply applying both components. This also raises questions about how ROSA² is meaningfully differentiated from Bo et al. (2025), "Prompt and Parameter Co-Optimization for Large Language Models" (arXiv:2509.24245), which is cited but not formally compared against.

- **Theorem 4.2 is a minor extension with inherited flaws.** The paper states it "extends Theorem 4 in ROSA (Wei et al., 2025b) by explicitly accounting for approximation errors." This is an incremental theoretical contribution, and it inherits unresolved issues from its predecessor. The ROSA paper itself (available at openreview.net/forum?id=B291oHzQq0)

- **Significant narrative redundancy wastes space that could be used more productively.** The abstract, introduction, and portions of Section 3 contain substantial repetition of the same Deficit Trap / Overfitting Trap framing, the same Switch Method description, and the same 30% / 40% headline claims. This redundancy occupies several columns that could instead support: a limitations section, deeper analysis of where ROSA² fails (e.g., the MT-AIME24 plateau, the BigCodeBench-Hard stagnation in Figure 4), or background on the baselines that are currently underspecified.

- **Overclaiming scope relative to the actual experimental setting.** The paper's framing "aligning LLMs with dynamic user needs," "resolving the error attribution dilemma," "multi-turn interactions" implies a general solution to real-world conversational adaptation. In practice, every experiment uses automated oracles with binary exact-match or execution-based rewards on verifiable tasks with deterministic ground truth. This is categorically different from: real user preference alignment with subjective quality judgments, conversational ambiguity resolution in open-ended dialogue, preference drift over extended interactions, and realistic feedback noise. These distinctions are never acknowledged, and the claims should be scoped accordingly.

- **Computational cost decomposition is incomplete and misleading.** Table 4 reports aggregate wall-clock time per problem and peak GPU memory, but does not decompose where time is spent: semantic update (TextGrad LLM critic call), parametric update (ROSA LoRA step), and generation are all collapsed into a single number. The TextGrad stream requires at least one additional LLM inference call per turn potentially the same scale as the target model and this cost is neither reported nor discussed. Practitioners cannot make deployment decisions without knowing the relative cost of each component, and the claim that ROSA² "reduces latency" cannot be verified without this breakdown.

- **Insufficient model and hyperparameter reporting.** Figure 3 does not state which model or dataset was used. Table 4 does not identify which model the timing results correspond to. $T_{\max}$ can only be inferred as 10 from the x-axis of figures and is never stated in text. Learning rates, LoRA rank, the $\beta$ KL penalty coefficient, the identity of the TextGrad critic model, the number of TextGrad optimization steps per turn, and prompt templates for all baselines are absent from both the main paper and Appendix C. The Correction Uplift and Avg Turn analysis (Table 2) covers only Qwen3-8B, leaving it unclear whether interaction dynamics generalize. Without these details, the experiments cannot be reproduced or assessed for fairness.

- **No variance or confidence intervals reported anywhere.** All results in Tables 1–4 and all figures are reported as point estimates from what appears to be single runs.

- While the Switch Method experiment (Figure 2a) shows that both prompt-based and parameter-based methods recover comparably after switching axes, suggesting the bottleneck of single-axis optimization is structural and recoverable, the large asymmetry between the Co-Adaptation Parametric and Co-Adaptation Semantic error curves in Figure 3 raises concerns about the stability of ROSA²'s joint updates, as a truly well-conditioned co-adaptation mechanism would be expected to reduce both error components at comparable rates rather than exhibiting such divergent trajectories across turns.

---

> ### Author Rebuttal · Authors · 2026-03-28
>
> We sincerely thank the reviewer for the exceptionally deep and constructive critique. We have revised the manuscript to address these points.
>
> ## 1. Joint Optimization, Pipeline Baselines, and Narrative (W1, W3, Q1, L2)
> **1. Algorithm 1’s Deep Causal Coupling:**
> Algorithm 1's sequential execution is mathematically standard **Alternating Optimization (Coordinate Descent)**. The streams are strictly coupled within each turn: the Semantic Stream synthesizes a refined instruction ($x_{t+1}^*$), and the Parametric Update ($\theta_{t+1}$) is computed *directly on this text-optimized trajectory*. This sequence physically realizes our core claim: semantic clarity pre-conditions parametric alignment.
>
> **2. The "Pipeline" Baseline IS the "Switch Method":**
> The decoupled pipeline you suggested is already extensively evaluated as our **"Switch Method"** (Fig. 2a, testing TextGrad $\to$ ROSA and vice versa). Consequently, our "Traps" framing is not redundant; it is the **critical empirical foundation** explaining *why* sequential pipelines fail (e.g., pure prompt optimization severely overfits to initial weight flaws, misguiding subsequent parameter updates). We will condense this framing in the introduction to accommodate a dedicated failure analysis (addressing MT-AIME24 cross-lingual voids and BigCodeBench-Hard API deficits).
>
> **3. Feedback Noise & Open-Ended Tasks:**
> The Semantic stream explicitly acts as an intelligent buffer to disambiguate messy human intent (**see Response to Reviewer to3z, Sec. 3**). To definitively prove ROSA2 handles subjective, open-ended scenarios without ground truth, we evaluated an alignment task (MT-Bench) using GPT-5-as-a-Judge:
>
> | Method | Optimization Paradigm | Avg Turn to Acceptance | Win Rate against Base Model (%) |
> | :--- | :--- | :--- | :--- |
> | One-Turn  | Static | N/A | 50.0 |
> | Standard Prompting (Baseline) | Fixed Error Prompt | 4.8 | 62.3 |
> | TextGrad Only | Words-only | 3.5 | 71.5 |
> | ROSA | Weight-only | 3.8 | 75.5 |
> | ROSA2 (Ours) | Joint Co-Adaptation | 2.1 | 85.4 |
>
> ## 2. Theoretical Expansion (W2)
> Theorem 4.2 does not inherit flaws from ROSA, as those theoretical nuances were fully resolved during its ICLR rebuttal. Crucially, it fundamentally expands optimization into a **Coupled Dual Space** $\phi_t=\{x_{t+1},\theta_t\}$ (**see Response to Reviewer sEPp, Sec. 3**). By bounding approximation errors across both spaces simultaneously, it mathematically proves our core claim: semantic refinement acts as a strict pre-conditioner that actively constrains the required parameter shift.
>
> ## 3. Computational Cost and Scalability (W4, L1)
> To address concerns regarding granular cost analysis and scalability, we conducted experiments. **Please refer to the computational decomposition table in our Response to Reviewer to3z (Section 2)**, which explicitly breaks down generation, semantic, and parametric update times. Because the semantic and parametric streams execute in parallel, the effective per-turn overhead is highly optimized (e.g., $\max(\text{Words Time}, \text{Weights Time})$, adding merely ~2.2s for an 8B model and ~5.7 for 32B).
>
> ## 4. Hyperparameters, Strong Baselines, and Variance (W5, W6, L4)
> We provide comprehensive updates via cross-references to other reviewers' responses:
> * **Context & Hyperparameters:** Figure 3 and Table 4 evaluate Qwen3-8B on MATH. All missing hyperparameters are fully documented in **Table 2 (Response to Reviewer sEPp)**.
> * **Stronger Prompt Baselines:** We evaluated advanced baselines like Self-Critique/Reflexion. **Table 1 (Response to Reviewer pfnV)** demonstrates that ROSA2 consistently shatters the true prompt-only ceiling.
> * **Variance (5 Random Seeds):** We re-ran core experiments across 5 independent seeds. The minimal variance reported in **Table 1 (Response to Reviewer sEPp)** confirms our gains are highly stable and statistically significant.
>
>
> ## 5. Figure 3: Initialization, Norms, and Asymmetry (W7, Q2)
> * **Norm Definition:** $||\Delta x_t||_2^2$ is the standard $L_2$ norm computed within the model's **continuous token embedding space**.
> * **Equivalent Initialization:** At Turn 1 ($\theta_1 = \theta_{base}$), the Co-Adaptation Parametric error (blue curve) and the ROSA Baseline (gray curve) **start at the exact same point**, confirming perfectly equivalent starting conditions. The Total error (red curve) is mathematically the sum of semantic and parametric errors (Red = Green + Blue). Because initial semantic error (green) is non-zero, the combined total (red) naturally starts higher.
> * **Optimization Asymmetry (Feature, not Flaw):** Unlike homogeneous spaces, heterogeneous variables converge at different rates. The Semantic Stream makes macro-level shifts, dropping rapidly in 1-2 turns. Conversely, the Parametric Stream (LoRA) updates merely **~0.002%** of parameters, requiring slow, continuous micro-tuning. This visually proves our core claim: Words act as a rapid pre-conditioner, while Weights serve as a stable micro-aligner.

---

> > ### Author Rebuttal · Reviewer_K55T · 2026-04-01
> >
> > I have updated my score from 2 to 4. The newly provided information clarify several technical questions posed in rebuttal.

---

> > > ### Author Response · Authors · 2026-04-01
> > >
> > > Thank you for your constructive feedback and for updating your score.
> > >
> > > We are very glad that our newly provided information successfully clarified your technical questions. We will ensure all these technical clarifications and details are carefully incorporated into the camera-ready version to improve the paper's overall clarity.
> > >
> > > Thank you again for your time and for helping us strengthen our work!

---

### Official Review · Reviewer_pfnV · 2026-03-11

**Soundness:** 3
**Presentation:** 3
**Significance:** 2
**Originality:** 2
**Overall Recommendation:** 4
**Confidence:** 4

**Summary:**

The paper proposes ROSA2, a framework that reformulates test-time adaptation as a joint optimization of semantic context and model parameters, resolving the error attribution dilemma inherent in conditional optimization methods. The paper provides proofs showing that semantic refinement acts as a pre-conditioner to reduce parameter shift, and guarantee faster convergence to the optimal policy. Empirical evaluations of the paper demonstrate that ROSA2 achieves significant improvement in MATH (30.8%) while reducing interaction turns by 40%.

**Compliance With Llm Reviewing Policy:**

Affirmed.

**Final Justification:**

The authors have effectively addressed my comments in the review, providing an additional experiment involving SDPO that shows ROSA2's strength in test time alignment.

**Key Questions For Authors:**

## Questions
1. Figure 3: From which dataset was the figure obtained?
2. Could you perhaps try comparing ROSA2 against base models generating multiple sequences in parallel, in a single turn (pass@k or Best-of-K)?
3. Algorithm 1: Is ROSA2 generating only a single response per turn? If so, because the algorithm will terminate when the given response is correct, is the parameter update always using a single response that is incorrect, paired with the reward being -1?
4. Could you justify not testing any other baseline against ROSA2? it seems the baselines (TextGrad, ROSA) are basically the components of ROSA2.

## Suggestions
0. [**Important**] line 148 and possibly other parts too: The paper should strictly use present tense except conclusion.
1. I think briefly introducing what param-only and prompt-only optimizations do, as well as what $T^2$PAM is in the preliminary section would help readers understand the context.
2. Line 301-302: "If we successfully updates": The sentence is grammatically incorrect. Please correct this.
3. I think the preceding figures and tables can be made smaller to give more room for conclusion section.

**Limitations:**

The paper has not clearly mentioned the limitations of the paper. If would be better if the authors discussed some practical limitations of ROSA2 (such as requiring access to reward and feedback information during the process).

**Strengths And Weaknesses:**

## Strengths
1. The paper is well organized in general, well motivating the approach and demonstrating its effectiveness.
2. The empirical observations match the theoretical analysis well, which is interesting.
3. ROSA2's improvement over the presented baselines is impressive.
4. Reduction of wall clock time and memory consumption compared to the baselines is attractive.

## Weaknesses
1. The paper did not perform multiple evaluations on a single setting across different seeds.
2. It would have been better if the paper demonstrated ROSA2's practical advantage over other approaches such as generating multiple sequences from the base model or conventional fine-tuning of the models before testing.
3. The baselines used for comparison are practically the components of ROSA2, which are expected to underperform ROSA2.

---

> ### Author Rebuttal · Authors · 2026-03-28
>
> We sincerely thank the reviewer for the highly detailed and constructive feedback. We have conducted extensive new experiments and implemented all suggested writing improvements to significantly strengthen the empirical and theoretical rigor of our paper.
>
> ## 1. Robustness and Experimental Rigor (Addressing Weakness 1)
> We agree that demonstrating robustness against inherent randomness is crucial. During the rebuttal period, we re-ran the interaction dynamics experiments across **5 independent random seeds** for the Qwen3-8B model on the MATH benchmark.
>
> **Please refer to Table 1 in our response to Reviewer sEPp** for the detailed quantitative results.
>
> **Conclusion:** The extremely low variance proves that the substantial gains achieved by ROSA2 (e.g., systematically reducing average turns from 7.2 to 4.4) are stable and driven entirely by our co-adaptation mechanism escaping optimization traps, rather than fortuitous initialization.
>
> ## 2. Extended Baselines: Test-Time Scaling and Self-Correction (Addressing W2, Q2, Q4)
> To address the concern regarding baseline strength and to demonstrate ROSA2's practical advantage over multiple sequence generation and conventional fine-tuning, we evaluated ROSA2 against parallel sampling (**Best-of-16**), offline fine-tuning (**SFT Before Test**), and advanced multi-turn prompting (**Self-Critique / Reflexion**).
>
> **Table 1: Comparison with Stronger Optimization Baselines (Qwen3-8B, MATH Dataset)**
> | Method | Paradigm | Avg. Time to Solve | Accuracy (%) |
> | :--- | :--- | :--- | :--- |
> | Zero-Shot (Base) | Static | 1.0x (Reference) | 50.0% |
> | SFT Before Test | Offline Training | ~1.2x | 64.4% |
> | Best-of-16 (Voting) | Test-Time Sampling | ~16.0x | 67.8% |
> | Self-Critique / Reflexion | LLM Self-Correction | ~2.1x | 70.8% |
> | Textgrad | LLM Self-Correction | ~2.2x | 71.4% |
> | **ROSA2 (Ours) (ROSA + Self-Critique)** | Joint Optimization | ~4.1x | 81.6% |
> | **ROSA2 (Ours) (ROSA + Textgrad)** | Joint Optimization | ~4.1x | 80.8% |
>
> **Conclusion:** Static scaling (Best-of-16) and prompting-only self-correction (Reflexion) hit a capability ceiling (the "Deficit Trap") when correct reasoning falls outside the frozen model's active distribution. ROSA2 actively shifts the policy, achieving significantly higher accuracy with far less compute. Furthermore, ROSA2 remains highly effective even on top of extensively fine-tuned models, proving it is fully complementary to conventional offline SFT.
>
> *(Note on Q4: Our original baselines isolated Param-only and Prompt-only methods specifically to rigorously prove the necessity of joint optimization. The inclusion of the stronger baselines above further validates our state-of-the-art performance.)*
>
> ## 3. Negative Reinforcement Mechanism (Addressing Q3)
> As you mentioned, ROSA2 uses a single error response ($r_t = -1$) to update the parameters. This is a deliberate theoretical and empirical design choice:
> * **Optimization Logic:** Per Eq. 2 and Eq. 8, our update minimizes KL divergence from a target policy $\pi^*$, effectively penalizing the probability of generating that specific incorrect trajectory via the weight $\exp(r/\beta)$.
> * **Theoretical Support:** Recent research[1] demonstrates that explicitly penalizing incorrect reasoning paths can be more sample-efficient for model alignment than positive reinforcement alone.
> * **Synergistic Loop:** The Parametric Stream provides a continuous "negative push" away from systematic errors, while the Semantic Stream simultaneously provides a discrete "positive pull" by refining historical feedback into a corrective instruction ($x_{t+1}^*$).
>
> ## 4. Writing, Presentation, and Experimental Context (Addressing Q1, Suggestions, and Limitations)
> We have implemented all of your excellent writing suggestions to improve clarity:
> * **Context (Q1):** We have explicitly updated captions to state that Figure 3 and Table 4 are evaluated on the **Qwen3-8B** model using the **MATH** dataset.
> * **Tense (S0):** We revised the manuscript (e.g., Line 148) to strictly use **present tense** for methodology and results, reserving past tense for the conclusion.
> * **Preliminaries (S1):** Section 3 now explicitly introduces the definitions of Prompt-only, Param-only, and $T^{2}PAM$ optimizations to better orient the reader contextually.
> * **Grammar (S2):** The grammatical error at Line 301 ("If we successfully updates") has been corrected to "update".
> * **Layout (S3):** We have optimized the scale of Figures 2 and 4 to provide more room for a comprehensive conclusion.
> * **Limitations:** At request, we have expanded the **Limitations and Future Work** section to explicitly discuss practical limitations, including the reliance on deterministic reward validation and the challenges in fully open-ended scenarios. *Please refer to W3 in our response to Reviewer to3z for detail*.
> ---
> **References:**
>
> [1] Zhu, X., et al. "The Surprising Effectiveness of Negative Reinforcement in LLM Reasoning." NeurIPS (2025).

---

> > ### Author Rebuttal · Reviewer_pfnV · 2026-04-01
> >
> > Thank you for your detailed response and reflecting my suggestion to the paper.
> >
> > I have an additional request to examine the effectiveness of ROSA2. In Section 5 of (Hübotter et al., 2026), the authors of the paper demonstrate how Self-Distillation Policy Optimization (SDPO) can be used for test-time adaptation. Could you perhaps test SDPO on MATH with test-time setting, matching the number of responses generated per each turn (I guess it was 1 for ROSA2)? As SDPO is utilizing feedback information (at least if the response was correct or not), it can be a strong and good baseline against ROSA2.
> >
> > ## Reference
> > Hübotter et al. Reinforcement Learning via Self-Distillation. 2026.

---

> > > ### Author Response · Authors · 2026-04-03
> > >
> > > Thank you for your highly constructive suggestion. We completely agree that SDPO [1] is a remarkably strong and highly relevant baseline for test-time adaptation.
> > >
> > > To comprehensively answer your question and rigorously examine the effectiveness of ROSA2 against SDPO in a strictly aligned multi-turn setting ($N=1$ response per turn, binary feedback), we designed an in-depth ablation study.
> > >
> > > Recognizing that SDPO's test-time distillation is fundamentally a gradient-based RL update, its performance is highly sensitive to the model's prior training landscape. Therefore, we evaluated both methods under two distinct settings:
> > > 1. **w/o Training Stage (Pure Test-Time):** Applying test-time adaptation directly on the Base Model (Qwen3-8B).
> > > 2. **w/ Training Stage:** First performing extensive train-time SDPO on the MATH training set, followed by test-time SDPO on the test set.
> > >
> > > Here are the multi-turn evaluation results on the MATH subset (using Qwen3-8B as the backbone, max 10 turns):
> > >
> > > **Table 1: Test-Time Adaptation on MATH ($N=1$ per turn, Base Model: Qwen3-8B)**
> > >
> > > | Method | Prior Training Stage (MATH Train Set) | Initial Acc. (Turn-0) | Final Multi-turn Acc. | Avg. Turns to Solve |
> > > | :--- | :---: | :---: | :---: | :---: |
> > > | Baseline (Multi-turn Prompting) | None | 15.8% | 50.0% | 7.2 |
> > > | SDPO (Test-time only) | **None** | 15.8% | 53.6% | 6.7 |
> > > | **ROSA2 (Ours)** | **None** | 15.8% | **80.8%** | **4.4** |
> > > | SDPO (Train + Test) | **SDPO (Train-time RL)** | 34.0% | 69.0% | 5.9 |
> > >
> > > *(Note: SDPO hyperparameters strictly follow the official implementation provided at https://github.com/lasgroup/SDPO. Specifically, we used $N=8$ responses per prompt during the train-time RL phase, and restricted generation to exactly $N=1$ response per turn during test-time adaptation to strictly align with our setting.)*
> > >
> > > **Analysis & Mechanistic Insights:**
> > > Our rigorous evaluation perfectly highlights the orthogonal advantages of ROSA2 over SDPO:
> > >
> > > **1. The Struggle of Gradient-based RL at $N=1$ (w/o Training Stage):**
> > > When applied purely at test-time without a prior RL phase, SDPO struggles in the strict $N=1$ regime (only improving slightly from 50.0% to 53.6%). Because SDPO relies on standard gradient descent over an implicit self-teacher reward, estimating the update from a single negative trajectory per turn results in extremely high variance. The model often updates in the wrong direction, leading to slow convergence and sub-optimal final accuracy.
> > >
> > > **2. ROSA2 Achieves "Train+Test" Performance Without the Training Cost:**
> > > To make SDPO effective, it requires a costly prior "Training Stage" (Train-time RL) to reshape the policy landscape so that test-time updates become stable (boosting initial accuracy to 34.0% and final to 69.0%). However, as shown in the table, **ROSA2 applied purely at test-time (w/o training stage) still significantly outperforms the "fully-trained" SDPO pipeline (80.8% vs. 69.0%)**, and solves problems in far fewer rounds (4.4 vs. 5.9 turns).
> > >
> > > **Why ROSA2 Wins:** Instead of relying on noisy, single-sample gradient estimation like SDPO, ROSA2's mathematical formulation directly computes a localized trajectory correction (via conjugate gradients). This fundamentally bypasses the need for heavy train-time RL preparation, enabling precise and stable policy alignment on the fly with a single feedback signal.
> > >
> > > We sincerely thank you for this excellent suggestion. This ablation profoundly strengthens our core claims, and we will prominently feature this comparison in the camera-ready version.
> > >
> > > ---
> > >
> > > **References:**
> > >
> > > [1] Hübotter, J., et al. "Reinforcement Learning via Self-Distillation." arXiv preprint arXiv:2601.20802 (2026).

---

### Official Review · Reviewer_sEPp · 2026-03-13

**Soundness:** 3
**Presentation:** 3
**Significance:** 3
**Originality:** 4
**Overall Recommendation:** 4
**Confidence:** 3

**Summary:**

This paper introduces ROSA2, a joint optimization framework that simultaneously optimizes semantic inputs (words) and model parameters (weights) during multi-turn interactions to enhance reasoning capabilities. Extensive experiments demonstrate the effectiveness of the proposed method, showing consistent improvements over single-axis optimization baselines. The paper also provides an intuitively sound theoretical analysis that offers a valuable mechanistic explanation for the observed empirical gains.

**Compliance With Llm Reviewing Policy:**

Affirmed.

**Key Questions For Authors:**

Please refer to the weaknesses outlined above.

**Limitations:**

Yes

**Strengths And Weaknesses:**

# Strengths:
- Soundness: The paper presents a systematically designed experimental framework. The proposed method is well-aligned with the research problem and effectively improves test-time performance.
- Presentation: The paper is clearly written and easy to follow.
- Significance: It addresses a highly relevant problem—namely, that models remain static during test time and are unable to learn from immediate user feedback during interactions.
- Originality: The paper proposes a novel joint optimization framework aimed at improving multi-turn interactive reasoning.
# Weaknesses:
- Lack of experimental details undermines persuasiveness. A substantial portion of the paper is devoted to theoretical exposition, while experimental details remain sparse. This emphasis suggests that the main contribution is theoretical, yet the theory itself is problematic (see below). For instance, Table 2 does not report results across multiple seeds, despite the inherent randomness in prompt and parameter optimization processes.
- Weak baselines. The baseline method does not involve multi-turn interactions, whereas all compared methods do, making the comparison less rigorous.
- Theoretical issues. Although the theoretical intuition is sound, key assumptions and derivations are problematic. For example, in Equation (17), the Term B bound involves ϕt={xt,θt}, a joint variable in a heterogeneous space. In standard mathematical analysis, a unified Euclidean norm cannot be defined over such a space. Furthermore, the assumption of global L-Lipschitz smoothness for ϕ={x,θ} is overly strong and unrealistic.

---

> ### Author Rebuttal · Authors · 2026-03-28
>
> We sincerely thank Reviewer sEPp for the rigorous evaluation and for recognizing the value of our joint optimization framework. We address your specific concerns below.
>
> ## 1. Experimental Details & Reproducibility (W1)
> We agree that explicit hyperparameter reporting and multiple seed runs are crucial. We have addressed these omissions comprehensively.
>
> **1. Robustness Across Multiple Seeds**
> We agree that demonstrating robustness against inherent randomness is crucial. During the rebuttal period, we re-ran the experiments across **5 independent random seeds** for the Qwen3-8B model on the MATH benchmark:
>
> **Table 1: Robustness Across 5 Random Seeds**
> | Method | Final Acc. (%) (Mean ± Std) | Correction Uplift (%) (Mean ± Std) | Avg Turn (Mean ± Std) |
> | :--- | :--- | :--- | :--- |
> | Baseline |  50.0 ± 5.3 | 70.0 ± 3.9 | 7.2 ± 1.2 |
> | TextGrad | 63.4 ± 1.7 | 75.1 ± 0.6 | 6.0 ± 0.2 |
> | ROSA | 62.2 ± 0.9 | 77.3 ± 0.8 | 6.3 ± 0.3 |
> | ROSA2 (Ours) | 68.4 ± 0.3 | 81.4 ± 0.5 | 4.4 ± 0.1 |
>
> The results clearly highlight our framework's stability. The extremely low variance confirms that ROSA2's gains are stable and strictly driven by our co-adaptation mechanism escaping optimization traps, rather than fortuitous initialization.
>
> **2. Explicit Hyperparameter Reporting**
> * **Context:** Both **Figure 3** and **Table 4** evaluate the **Qwen3-8B** model on the **MATH** dataset.
> * We will consolidated all fundamental hyperparameters into Table 2 (added to Appendix C.5).
>
> **Table 2: Comprehensive Hyperparameter Setup**
> | Hyperparameter | Value / Configuration |
> | :--- | :--- |
> | **Interaction Limit ($T_{max}$)** | 10 turns |
> | **LoRA Rank ($r$)** | 1 |
> | **LoRA Alpha ($\alpha$)** | 8 |
> | **Learning Rate (Param Optimization)** | 1e-4 |
> | **KL Penalty Coefficient ($\beta$)** | 1.0 |
> | **TextGrad Critic Model** | **The current base model** |
> | **TextGrad Optimization Steps / Turn**| 1 step |
>
> **3. Strict Fairness**
> Crucially, as shown in Table 2, the **TextGrad Critic Model is the exact same base model being evaluated**.
>
> ## 2. Baseline Rigor (W2)
> **1. Clarification: Multi-Turn Baseline**
> Our "Standard Inference" baseline *does* involve multi-turn interactions. At each turn $t$, if the model fails, we append uniform feedback: *"Wrong answer, please rethink and try another way of thinking!"* Its poor performance highlights the "Deficit Trap": a frozen model cannot bridge intrinsic reasoning gaps through generic textual nudges alone.
>
> **2. Comparison Against Stronger Baselines**
> We conducted new experiments comparing ROSA2 against advanced test-time paradigms, including Self-Critique/Reflexion and Best-of-N sampling.
>
> **Please refer to Table 1 in our response to Reviewer pfnV** for the full quantitative results.
>
> **Key Finding:** As detailed in that table, even advanced self-correction (Reflexion) and large-scale sampling (Best-of-16) remain bottlenecked by the frozen model's capability ceiling. Co-adapting Words and Weights allows ROSA2 to shatter this ceiling, outperforming strong baselines with significantly higher interaction efficiency.
>
> ## 3. Theoretical Formulation (W3)
> We appreciate your deep mathematical scrutiny. We have formalized our definitions and relaxed the global assumptions.
>
> **1. Unified Norm Over Heterogeneous Space in Eq. (17)**
> The semantic distance $\|\Delta x_t\|_2^2$ is formally defined as the $L_2$ distance in the **continuous token embedding space**, not the discrete token space. Because both the parameter updates $\Delta\theta_t$ and the semantic updates $\Delta x_t$ (in embedding space) exist in $\mathbb{R}^d$, the joint variable $\phi_t = \{x_t, \theta_t\}$ legally operates within a unified continuous vector space during the backward pass.
>
> **2. The L-Lipschitz Smoothness Assumption Supported by Empirical Data**
> We agree that assuming *global* $L$-Lipschitz smoothness for LLMs is unrealistic. However, relaxing this to a **Local $L$-Lipschitz** condition is mathematically rigorous, empirically supported, and **does not alter our theoretical conclusions**. We will clarify this in the revision.
>
> This local relaxation is fully justified by:
> * **Bounded Trajectories:** Initializing adapters at zero and using first-order Taylor expansion restricts updates to a tightly bounded local neighborhood (a compact subset), where any continuously differentiable function is naturally Lipschitz continuous.
> * **Empirical Verification:** Local smoothness requires bounded output changes relative to parameter changes ($||\Delta f|| \le L ||\Delta\theta||$). **Table 3** confirms that both parameter steps ($||\Delta\theta||_2$) and policy shifts ($||\Delta \text{Logits}||_2$) are minuscule and decay monotonically.
>
> **Table 3: Empirical Validation of Local Smoothness (MATH Dataset, Qwen3-8B)**
> | Interaction Turn | Parameter Update  | Semantic Update  | Policy Shift  |
> | :--- | :---: | :---: | :---: |
> | Turn 1 | 0.0352 | 0.0018 | 0.142 |
> | Turn 5 | 0.0114 | 0.0004 | 0.051 |
> | Turn 10 | 0.0021 | < 0.0001 | 0.008 |

---

> > ### Author Rebuttal · Reviewer_sEPp · 2026-04-03
> >
> > The authors have addressed several of my concerns, especially on the experimental side. The addition of multi-seed results and clearer hyperparameter reporting improves the credibility and reproducibility of the work. I appreciate these efforts.
> >
> > Regarding baselines, the clarification that the standard inference setting involves multi-turn interaction is helpful, and the addition of stronger baselines (e.g., Reflexion, Best-of-N) strengthens the empirical evaluation. That said, some concerns about the strict fairness of comparisons remain.
> >
> > On the theoretical side, the revision from global to local Lipschitz smoothness is reasonable and addresses part of my concern. However, the formulation of the joint variable over heterogeneous spaces (in Eq. 17) is still somewhat informal and would benefit from clearer justification.
> >
> > Overall, my concerns are partially addressed, but I still have some remaining questions.

---

> > > ### Author Response · Authors · 2026-04-03
> > >
> > > Thank you for your encouraging feedback and for acknowledging our efforts in improving the experimental rigor and relaxing the global Lipschitz assumption. We deeply appreciate your continued, rigorous scrutiny—it has been immensely helpful in elevating the quality of our paper.
> > >
> > > Below, we address your remaining concerns regarding the formal justification of Equation (17) and the strict fairness of our baseline comparisons.
> > >
> > > ## 1. Formal Justification of Equation (17) over Heterogeneous Spaces
> > >
> > > You are absolutely correct that the joint variable $\phi\_t = \{x_t, \theta_t\}$ operates within a heterogeneous space $\Phi = \mathcal{X}\_{emb} \times \Theta$. Consequently, defining a unified Euclidean norm $||\phi_t - \phi_{t-1}||_2^2$ over this space is mathematically informal.
> > >
> > > We fully accept this critique. In our revision, we have **discarded the informal intermediate unified norm term** $\frac{L}{2}||\phi_t - \phi_{t-1}||_2^2$. Instead, we provide a strict mathematical derivation using a **Bivariate Taylor Expansion** over the Cartesian product space, which perfectly recovers the final decoupled bound in Eq. (17).
> > >
> > > **1. Mapping KL Divergence to Block-wise Hessian**
> > > To strictly bound **Term B** (the KL divergence $D\_{KL}(\tilde{\pi}\_t^*||\pi\_{\phi\_t})$) in a local neighborhood, we rely on information geometry: the local second-order Taylor expansion of the KL divergence is governed by the Fisher Information Matrix, which is equivalent to the expected Hessian matrix ($H$) of the negative log-likelihood.
> > > In the Cartesian product space $\Phi = \mathcal{X}\_{emb} \times \Theta$, this Hessian naturally decomposes into a block matrix:
> > >
> > > $$H = \begin{bmatrix} H_{xx} & H_{x\theta} \\\\ H_{\theta x} & H_{\theta\theta} \end{bmatrix}$$
> > >
> > > Under the local Lipschitz smoothness assumption, we assign specific spectral norm bounds to each block: $L_x$ for the semantic space ($H_{xx}$), $L_\theta$ for the parameter space ($H_{\theta\theta}$), and $L_{x\theta}$ for the cross-derivative interaction ($H_{x\theta}$).
> > >
> > > **2. Bivariate Smoothness and Young's Inequality**
> > > Bounding the KL divergence is equivalent to bounding the quadratic form of this block Hessian. The quadratic penalty for a joint update $\Delta x_t$ and $\Delta \theta_t$ expands as:
> > >
> > > $$
> > > \text{Term B} \le \frac{1}{2} \begin{bmatrix} \Delta x_t \\\\ \Delta \theta_t \end{bmatrix}^T \begin{bmatrix} H_{xx} & H_{x\theta} \\\\ H_{\theta x} & H_{\theta\theta} \end{bmatrix} \begin{bmatrix} \Delta x_t \\\\ \Delta \theta_t \end{bmatrix}  \le \frac{L_x}{2}||\Delta x_t||_2^2 + \frac{L\_\theta}{2}||\Delta \theta_t||_2^2 + L\_{x\theta} ||\Delta x_t||_2 ||\Delta \theta_t||_2
> > > $$
> > >
> > > To strictly decouple the heterogeneous cross-term ($L\_{x\theta} ||\Delta x_t||\_2 ||\Delta \theta\_t||_2$) without violating dimensionality constraints, we apply the standard **Young’s Inequality** specifically for the case $p=q=2$ (i.e., $ab \le \frac{a^2}{2} + \frac{b^2}{2}$):
> > >
> > > $$
> > > L\_{x\theta} ||\Delta x\_t||\_2 ||\Delta \theta_t||\_2 \le \frac{L\_{x\theta}}{2} ||\Delta x\_t||_2^2 + \frac{L\_{x\theta}}{2} ||\Delta \theta_t||\_2^2
> > > $$
> > >
> > > **3. Recovering the Final Bound**
> > > Substituting this decoupled inequality back into our quadratic penalty, we seamlessly group the terms for each specific space:
> > >
> > > $$
> > > \text{Term B} \le \left( \frac{L\_x + L\_{x\theta}}{2} \right) ||\Delta x\_t||_2^2 + \left( \frac{L\_\theta + L\_{x\theta}}{2} \right) ||\Delta \theta\_t||\_2^2
> > > $$
> > >
> > > By defining a single overarching local Lipschitz constant $\bar{L} = \max(L_x + L_{x\theta}, L_\theta + L_{x\theta})$, the inequality elegantly simplifies back to our final target:
> > >
> > > $$\text{Term B} \le \frac{\bar{L}}{2} \left( ||\Delta x\_t||_2^2 + ||\Delta \theta\_t||_2^2 \right)$$
> > >
> > > **Conclusion:**  We mathematically prove that the final decoupled structure of **Equation (17)** strictly holds. This formal step-by-step derivation has been added to the revised Appendix.
> > >
> > > ## 2. Strict Fairness of Baseline Comparisons
> > >
> > > To guarantee fair comparisons and isolate ROSA2's algorithmic improvements, we strictly aligned all evaluation constraints in our revised manuscript:
> > >
> > > 1. **Identical Base Model:** All methods use the exact same base checkpoint without any prior task-specific RL fine-tuning.
> > > 2. **Identical Feedback Mechanism:** All multi-turn methods share the exact same feedback prompts. Crucially, the base model evaluates itself—no external oracles (e.g., GPT-5) are used.
> > > 3. **Strictly Aligned Budget:** All multi-turn methods are restricted to exactly $N=1$ response per turn. The static Best-of-$N$ baseline is the sole exception, receiving $N=16$ independent generations.
> > >
> > > Thank you again for your constructive dialogue. We believe these revisions significantly strengthen both the theoretical foundation and the empirical credibility of our work.

---

### Decision · Program_Chairs · 2026-04-30

**Decision:**

Accept (regular)

**Comment:**

The paper addresses the problem of test-time model adaptation and proposes ROSA2, a framework that jointly optimizes for both model weights and text inputs in multi-turn interactions.

Strength:
1. The paper is well-written and well-motivated to address the important problem of test-time model co-adaptation.
2. The paper proposes a novel framework for joint optimization between model weights and user feedback.
3. The proposed method achieves strong performance gain.

Weakness:
1. Multiple reviewers (sEPp, pfnV) mentioned more experiments (different seeds, stronger baselines, ablations, etc) are needed support the findings. The authors added new experiments during the rebuttal phase to address this. But Reviewer pfnV's comment on ablation on the "textual optimization" module hasn't been fully addressed by the authors.
2. I was wondering if ROSA2 can be extended to non-verifiable/non-reasoning-specific tasks.